# Exploiting Correlated Auxiliary Feedback in Parameterized Bandits

**Arun Verma**     **Zhongxiang Dai**     **Yao Shu**     **Bryan Kian Hsiang Low**
Department of Computer Science, National University of Singapore, Republic of Singapore
{arun, daizhongxiang, shuyao, lowkh}@comp.nus.edu.sg

## Abstract

We study a novel variant of the parameterized bandits problem in which the learner can observe additional auxiliary feedback that is correlated with the observed reward. The auxiliary feedback is readily available in many real-life applications, e.g., an online platform that wants to recommend the best-rated services to its users can observe the user's rating of service (rewards) and collect additional information like service delivery time (auxiliary feedback). In this paper, we first develop a method that exploits auxiliary feedback to build a reward estimator with tight confidence bounds, leading to a smaller regret. We then characterize the regret reduction in terms of the correlation coefficient between reward and its auxiliary feedback. Experimental results in different settings also verify the performance gain achieved by our proposed method.

## 1   Introduction

Parameterized bandits (Slivkins, 2019; Lattimore and Szepesvári, 2020) have many real-life applications in online recommendation, advertising, web search, and e-commerce. In this bandit problem, a learner selects an action and receives a reward for the selected action. Due to the large (or infinite) number of actions, the mean reward of each action is assumed to be parameterized by an unknown function, e.g., linear (Li et al., 2010; Chu et al., 2011; Abbasi-Yadkori et al., 2011; Agrawal and Goyal, 2013), GLM (Filippi et al., 2010; Li et al., 2017; Jun et al., 2017), and non-linear (Valko et al., 2013; Chowdhury and Gopalan, 2017). The learner aims to learn the best action as quickly as possible. However, it depends on the tightness of confidence bounds of function that correlate action with the reward. The learner exploits any available information like side information (i.e., information available to the learner before selecting an action, e.g., contexts) (Li et al., 2010; Agrawal and Goyal, 2013; Li et al., 2017) and side observations (i.e., information about other actions, e.g., graph feedback) (Alon et al., 2015; Wu et al., 2015) to make confidence bounds as tight as possible. This paper considers another type of additional information (correlated with the reward) that a learner can observe with reward for the selected action, which we call *auxiliary feedback*.

The auxiliary feedback is readily available in many real-life applications. For example, consider an online food delivery platform that wishes to recommend the best restaurants (actions) to its users. After receiving food, the platform observes user ratings (rewards) for the order and can collect additional information like food delivery time (auxiliary feedback). Since the restaurant's rating also depends on overall food delivery time, one can expect it to be correlated with the user rating. The platform can estimate or know the average delivery time for a given order from historical data. Similar scenarios arise in recommending the best cab to users (auxiliary information can be the cab's distance from the rider or driver's response to ride request), e-commerce platforms choosing top sellers to buyers (auxiliary information can be seller's response time for order confirmation and delivery), queuing network (Lavenberg and Welch, 1981; Lavenberg et al., 1982), jobs scheduler (Verma and Hanawal, 2021), and many more. Therefore, the following question naturally arises:
***How to exploit correlated auxiliary feedback to improve the performance of a bandit algorithm?***

37th Conference on Neural Information Processing Systems (NeurIPS 2023).

One possible method is to use auxiliary feedback in the form of control variates (Lavenberg et al., 1982; Nelson, 1990) for the observed reward. A control variate represents any random variable (auxiliary feedback) with a known mean that is correlated with the random variable of interest (reward). Several works (Kreutzer et al., 2017; Sutton and Barto, 2018; Vlassis et al., 2021; Verma and Hanawal, 2021) have used control variates to estimate the mean reward estimator with smaller variance, leading to tight confidence bounds and hence better performance. The closest work to our setting is Verma and Hanawal (2021). However, it only focuses on the non-parameterized setting and assumes a finite number of actions. We thus consider a more general bandit setting with a large (or even infinite) number of actions and allow an unknown function to parameterize auxiliary feedback.

Motivated by control variate theory (Nelson, 1990), we first introduce *hybrid reward*, which combines the reward and its auxiliary feedback in such a way that hybrid reward is an unbiased reward estimator with smaller variance than the observed reward. However, the optimal combination of reward and its auxiliary feedback requires knowledge of the covariance matrix among auxiliary feedback and covariance between reward and its auxiliary feedback, which may not be available in practice. Since the reward and its auxiliary feedback are functions of the selected action, existing control variate results can not be useful to our sequential setting. Naturally, we face the question of *how to combine reward and its auxiliary feedback efficiently using available information.* To answer this, we extend control variate theory results to the problems where known functions can parameterize control variates (in Section 3) and then extend to setting where unknown functions parameterize control variates (in Section 4). These contributions are themselves of independent interest in control variate theory.

Equipped with these results, we show that the variance of hybrid rewards is smaller than observed rewards (Theorem 1 and Theorem 3). We then propose a method that uses hybrid rewards instead of observed rewards for estimating reward function, resulting in tight confidence bounds and, hence, lower regret. We introduce the notion of *Auxiliary Feedback Compatible* (AFC) bandit algorithm. An AFC bandit algorithm can use hybrid rewards instead of only observed rewards. We prove that the expected instantaneous regret of any AFC bandit algorithm using hybrid rewards is smaller by a factor of $O((1-\rho^2)^{\frac{1}{2}})$ compared to the same AFC bandit algorithm using only observed rewards, where $\rho$ is the correlation coefficient of the reward and its auxiliary feedback (Theorem 2 and Theorem 4). Our experimental results in different settings also verify our theoretical results (in Section 5).

## 1.1 Related work

Several prior works use additional information to improve the performance of bandit algorithms. In the following, we discuss how auxiliary feedback differs from side information and side observation.

**Side Information:** Several works use context as side information to select the best action to play. This line of work is popularly known as contextual bandits (Li et al., 2010; Chu et al., 2011; Agrawal and Goyal, 2013; Li et al., 2017). Here, the mean reward of each arm is a function of context and is often parameterized, e.g., linear (Li et al., 2010; Chu et al., 2011; Agrawal and Goyal, 2013), GLM (Li et al., 2017), and non-linear (Valko et al., 2013). These contexts are assumed to be observed *before* an action is taken. However, we consider a problem where additional information is correlated with rewards that can only be observed *after* selecting the action.

**Side Observations:** Several works consider the different side observations settings in the literature, e.g., stochastic (Caron et al., 2012), adversarial (Mannor and Shamir, 2011; Kocák et al., 2014), graph feedback (Alon et al., 2015; Wu et al., 2015; Alon et al., 2017), and cascading feedback (Verma et al., 2019, 2020a,b). Side observations represent the additional information available about actions that the learner does *not select*. Auxiliary feedback is different from side information as it is available only for *selected* action and provides more information about the reward of that action.

**Auxiliary Feedback:** We use auxiliary feedback as control variates, which are used extensively for variance reduction in Monte-Carlo simulation of complex systems (Lavenberg and Welch, 1981; Lavenberg et al., 1982; James, 1985; Nelson, 1989, 1990; Botev and Ridder, 2017; Chen and Ghahramani, 2016). Recent works (Kreutzer et al., 2017; Vlassis et al., 2021; Verma and Hanawal, 2021) and (Sutton and Barto, 2018, Chapter 7.4) have exploited the availability of these control variates to build estimators with smaller variance and develop algorithms that have better performance guarantees. The closest work to our setting is Verma and Hanawal (2021). However, they only consider a non-parameterized setting with a finite number of actions. We thus consider a more general bandit setting with large (infinite) actions and allow a function to parameterize auxiliary feedback.

## 2 Problem setting

We consider a novel parameterized bandits problem in which the learner can observe auxiliary feedback correlated with the observed reward. In this problem, a learner has been given an action set, denoted by $\mathcal{X} \subset \mathbb{R}^d$ where $d \geq 1$. At the beginning of round $t$, the learner selects an action $x_t$ from action set $\mathcal{X}$. Then, the environment generates a stochastic reward $y_t \doteq f(x_t) + \varepsilon_t$ for the selected action $x_t$, where $f : \mathbb{R}^d \to \mathbb{R}$ is an unknown reward function and $\varepsilon_t$ is a zero-mean Gaussian noise with variance $\sigma^2$. Apart from the stochastic reward $y_t$, the environment generates $q$ of auxiliary feedback. The $i$-th auxiliary feedback is denoted by $w_{t,i} \doteq g_i(x_t) + \varepsilon_{t,i}^w$, where $g_i : \mathbb{R}^d \to \mathbb{R}$ and $\varepsilon_{t,i}^w$ is a zero-mean Gaussian noise with variance $\sigma_{w,i}^2$. The multiple correlation coefficient of reward and its auxiliary feedback is denoted by $\rho$ and assumed to be the same across all actions.

The optimal action (denoted by $x^\star$) has the maximum function value, i.e., $x^\star \in \mathrm{argmax}_{x \in \mathcal{X}} f(x)$. After selecting an action $x_t$, the learner incurs a penalty (or *instantaneous regret*) $r_t$, where $r_t \doteq f(x^\star) - f(x_t)$. Since the optimal action is unknown, we sequentially estimate the reward function using available information on rewards and associated auxiliary feedback for the selected actions and then use it for choosing the action in the following round. Our goal is to learn a sequential policy that selects actions such that the total penalty (or *regret*) incurred by the learner is as minimum as possible. After $T$ rounds, the regret of a sequential policy $\pi$ that selects action $x_t$ in the round $t$ is given by

$$\mathfrak{R}_T(\pi) \doteq \sum_{t=1}^{T} r_t = \sum_{t=1}^{T} \left( f(x^\star) - f(x_t) \right). \tag{1}$$

A policy $\pi$ is a good policy if it has sub-linear regret, i.e., $\lim_{T \to \infty} \mathfrak{R}_T(\pi)/T = 0$. This implies that the policy $\pi$ will eventually learn to recommend the best action.

## 3 Known auxiliary feedback functions

We first focus on a simple case where all auxiliary feedback functions are assumed to be known. This assumption is not very strict in many applications as the learner can construct auxiliary feedback such that its mean value is known beforehand (see Kreutzer et al. (2017), Vlassis et al. (2021), and Chapter 12.9 of Sutton and Barto (2018) for such examples). When auxiliary feedback functions are unknown, we can estimate them using historical data or additional samples of auxiliary feedback. However, it will have some penalty in the performance (more details are in Section 4 and Section 5).

The first challenge we face is *how to exploit auxiliary feedback to get a better reward function estimator*. To resolve this, we extend control variate theory (Nelson, 1990) to the problems where a function can parameterize control variates. This new contribution is itself of independent interest.

### 3.1 Control variate

Let $\mu$ be the unknown variable that needs to be estimated and $y$ be its unbiased estimator, i.e., $\mathbb{E}[y] = \mu$. Any random variable $w$ with a known mean value ($\omega$) can be treated as a control variate for $y$ if it is correlated with $y$. The control variate method (Nelson, 1990) exploits errors in estimates of known random variables to reduce the estimator's variance for the unknown random variable. This method works as follows. For any choice of a coefficient $\beta$, define a new random variable as $z \doteq y - \beta(w - \omega)$. Note that $z$ is also an unbiased estimator of $\mu$ (i.e., $\mathbb{E}[z] = \mu$) as

$$\mathbb{E}[z] = \mathbb{E}[y] - \beta \mathbb{E}[(w - \omega)] = \mu - \beta(\mathbb{E}[w] - \omega) = \mu - \beta(\omega - \omega) = \mu.$$

Using properties of variance and covariance, the variance of $z$ is given by

$$\mathbb{V}(z) = \mathbb{V}(y) + \beta^2 \mathbb{V}(w) - 2\beta \mathrm{Cov}(y, w).$$

The variance of $z$ is minimized by setting $\beta$ to $\beta^\star = \mathrm{Cov}(y, w)/\mathbb{V}(w)$ and the minimum value is $(1 - \rho^2)\mathbb{V}(y)$, where $\rho = \mathrm{Cov}(y, w)/\sqrt{\mathbb{V}(w)\mathbb{V}(y)}$ is the correlation coefficient of $y$ and $w$. We exploit this variance reduction to design a reward function estimator with tight confidence bounds.

### 3.2 Auxiliary feedback as control variates

Since the auxiliary feedback functions are known, we define a new variable using the reward sample and its auxiliary feedback. We refer to this variable as *'hybrid reward.'* The hybrid reward definition

is motivated by the control variate method, except the control variate is parameterized by function in our setting. As $w_{s,i}$ is the $i^{\text{th}}$ auxiliary feedback observed with reward $y_s$, the *hybrid reward* for reward $(y_s)$ with its $q$ auxiliary feedback $\{w_{s,i}\}_{i=1}^{q}$ is defined by

$$z_{s,q} \doteq y_s - \sum_{i=1}^{q} \beta_i(w_{s,i} - g_i(x_s)) = y_s - (\boldsymbol{w}_s - \boldsymbol{g}_s)\boldsymbol{\beta}, \tag{2}$$

where $\boldsymbol{w}_s = (w_{s,1}, \ldots, w_{s,q})$, $\boldsymbol{g}_s = (g_1(x_s), \ldots, g_q(x_s))$, and $\boldsymbol{\beta} = (\beta_1, \ldots, \beta_q)^{\top}$. Let $\Sigma_{\boldsymbol{ww}} \in \mathbb{R}^{q \times q}$ be the covariance matrix among auxiliary feedback and $\sigma_{\boldsymbol{yw}} \in \mathbb{R}^{q \times 1}$ be the vector of covariance between the reward and each of its auxiliary feedback. Then, the variance of $z_{s,q}$ is minimized by setting the coefficient vector $\boldsymbol{\beta}$ to $\boldsymbol{\beta}^{\star} = \Sigma_{\boldsymbol{ww}}^{-1}\sigma_{\boldsymbol{yw}}$, and the minimum value is $(1 - \rho^2)\sigma^2$, where $\rho^2 = \sigma_{\boldsymbol{yw}}^{\top}\Sigma_{\boldsymbol{ww}}^{-1}\sigma_{\boldsymbol{yw}}/\sigma^2$ is the multiple correlation coefficient of reward and its auxiliary feedback.

However, $\Sigma_{\boldsymbol{ww}}$ and $\sigma_{\boldsymbol{yw}}$ can be unknown in practice and need to be estimated to get the best estimate for $\boldsymbol{\beta}^{\star}$ to achieve maximum variance reduction. In our following result, we drive the best linear unbiased estimator of $\boldsymbol{\beta}$ (i.e., $\hat{\boldsymbol{\beta}}_t$) using $t$ observations of rewards and their auxiliary feedback.

**Lemma 1.** *Let $t > q + 2 \in \mathbb{N}$ and $f_t$ be the estimate of function $f$ which uses all information available at the end of round $t$, i.e., $\{x_s, y_s, \boldsymbol{w}_s\}_{s=1}^{t}$. Then, the best linear unbiased estimator of $\boldsymbol{\beta}^{\star}$ is*

$$\hat{\boldsymbol{\beta}}_t \doteq (\boldsymbol{W}_t^{\top}\boldsymbol{W}_t)^{-1}\boldsymbol{W}_t^{\top}\boldsymbol{Y}_t,$$

*where $\boldsymbol{W}_t$ is a $t \times q$ matrix whose $s^{\text{th}}$ row is $(\boldsymbol{w}_s - \boldsymbol{g}_s)$ and $\boldsymbol{Y}_t = (y_1 - f_t(x_1), \ldots, y_t - f_t(x_t))$.*

The proof follows after doing some manipulations in Eq. (2) and then using results from linear regression theory. The detailed proof of Lemma 1 and all other missing proofs are given in the supplementary material. After having a new observation of reward and its auxiliary feedback, the best linear unbiased estimator of $\boldsymbol{\beta}^{\star}$ is re-estimated. If $\Sigma_{\boldsymbol{ww}}$ or $\sigma_{\boldsymbol{yw}}$ are known, we can directly use them to estimate $\boldsymbol{\beta}^{\star}$ by replacing $\boldsymbol{W}_t^{\top}\boldsymbol{W}_t$ with $\Sigma_{\boldsymbol{ww}}$ and $\boldsymbol{W}_t^{\top}\boldsymbol{Y}_t$ with $\sigma_{\boldsymbol{yw}}$ in Lemma 1. The following result describes the properties of the hybrid reward when $\boldsymbol{\beta}^{\star}$ is replaced by $\hat{\boldsymbol{\beta}}_t$ in Eq. (2).

**Theorem 1.** *Let $t > q + 2 \in \mathbb{N}$. If $\hat{\boldsymbol{\beta}}_t$ as defined in Lemma 1 is used to compute hybrid reward $z_{s,q}$ for any $s \leq t \in \mathbb{N}$, then $\mathbb{E}[z_{s,q}] = f(x_s)$ and $\mathbb{V}(z_{s,q}) = \left(1 + \frac{q}{t-q-2}\right)(1 - \rho^2)\sigma^2$.*

The key takeaways from Theorem 1 are as follows. First, the hybrid reward using $\hat{\boldsymbol{\beta}}_t$ is an unbiased estimator of reward function and hence, we can still use it to estimate the reward function $f$. Second, there is a less reduction in variance (i.e., by a factor $(t-2)/(t-q-2)$ of maximum possible variance reduction) when $\hat{\boldsymbol{\beta}}_t$ is used for constructing hybrid reward in Eq. (2).

The variance reduction (given in Theorem 1) depends on the auxiliary feedback via two terms: $\frac{t-2}{t-q-2}$ and $\rho^2$ (defined in Line 142). Setting $q = 1$ gives the minimum value for $\frac{t-2}{t-q-2}$, but $\rho^2$ for $q = 1$ will also be small as it only considers one auxiliary feedback, and hence maximum variance reduction will not be achieved. As we increase the number of auxiliary feedback, $\rho^2$ increases, leading to more variance reduction. However, the term $\frac{t-2}{t-q-2}$ increases at the same time, which can negate the variance reduction achieved by smaller $\rho^2$. Hence, keeping the number of auxiliary feedback used for hybrid reward small is important. A simple method for selecting a subset of auxiliary feedback (Lavenberg et al., 1982) works as follows: select the auxiliary feedback whose sample correlation coefficient with reward is the largest. Then, select the next auxiliary feedback whose sample partial correlation coefficient with reward was the largest given the first auxiliary feedback selected. Keep repeating the process until there is a variance reduction using additional auxiliary feedback.

### 3.3 Linear bandits with known auxiliary feedback functions

To highlight the main ideas, we restrict to the linear bandit setting in which the reward and auxiliary feedback functions are linear. In this setting, a learner selects an action $x_t$ and observes a reward $y_t = x_t^{\top}\theta^{\star} + \varepsilon_t$, where $\theta^{\star} \in \mathbb{R}^d$ ($d \geq 1$) is unknown and $\varepsilon_t$ is the zero-mean Gaussian noise with known variance $\sigma^2$. The learner also observes $q$ auxiliary feedback, where $i$-th auxiliary feedback is denoted by $w_{t,i} = x_t^{\top}\theta_{w,i}^{\star} + \varepsilon_{t,i}^{w}$. Here, $\theta_{w,i}^{\star} \in \mathbb{R}^d$ is known and $\varepsilon_{t,i}^{w}$ is the zero-mean Gaussian noise with unknown variance $\sigma_{w,i}^2$. Our goal is to learn a policy that minimizes regret as defined in Eq. (1). We later extend our method for non-linear reward and auxiliary feedback functions in Section 4.3.

Let $I$ be the $d \times d$ identity matrix, $V_t \doteq \sum_{s=1}^{t-1} x_s x_s^\top$, and $\overline{V}_t \doteq V_t + \lambda I$, where $\lambda > 0$ is the regularization parameter that ensures matrix $\overline{V}_t$ is a positive definite matrix. The notation $\|x\|_A$ denotes the weighted $l_2$-norm of vector $x \in \mathbb{R}^d$ with respect to a positive definite matrix $A \in \mathbb{R}^{d \times d}$.

As shown in Theorem 1, the hybrid rewards is an unbiased reward estimator with a smaller variance than observed rewards. Thus, hybrid rewards lead to tighter confidence bounds for parameter $\theta^\star$ than observed rewards. We propose a simple but effective method to exploit correlated auxiliary feedback, i.e., using hybrid rewards to estimate reward function instead of observed rewards.

Using this method, we adapt the well-known linear bandit algorithm OFUL (Abbasi-Yadkori et al., 2011) to our setting and named this algorithm OFUL-AF. This algorithm works as follows. It takes $\lambda > 0$ as input and then initializes $\overline{V}_1 = \lambda I$ and sets $\hat{\theta}_1^z = 0_{\mathbb{R}^d}$ as initial estimate of parameter $\theta^\star$. The superscript '$z$' in $\hat{\theta}_1^z$ implies that hybrid rewards are used for estimating $\theta^\star$. At the beginning of round $t$, the algorithm selects an action $x_t$ that maximizes the upper confidence bound of the action's reward, which is a sum of the estimated reward for the action ($x^\top \hat{\theta}_t^z$) and a confidence bonus $\alpha_t^\sigma \|x\|_{\overline{V}_t^{-1}}$. In the confidence bonus, the first term ($\alpha_t^\sigma$) is a slowly increasing function in $t$ whose value is given in Theorem 2, and the second term ($\|x\|_{\overline{V}_t^{-1}}$) decreases to zero as $t$ increases.

---

**OFUL-AF** Algorithm for Linear Bandits with Auxiliary Feedback

---

1: **Input:** $\lambda > 0$
2: **Initialization:** $\overline{V}_1 = \lambda I$ and $\hat{\theta}_1^z = 0_{\mathbb{R}^d}$
3: **for** $t = 1, 2, \ldots$ **do**
4:      Select action $x_t = \operatorname{argmax}_{x \in \mathcal{X}} \left( x^\top \hat{\theta}_t^z + \sigma \alpha_t \|x\|_{\overline{V}_t^{-1}} \right)$
5:      Observe reward $y_t$ and its auxiliary feedback $\{w_{t,i}\}_{i=1}^q$
6:      If $t > q + 2$, compute upper bound of hybrid reward's sample variance ($\bar{\nu}_{z,t}$) or else $\bar{\nu}_{z,t} = \sigma^2$
7:      If $t \leq q + 2$ or $\bar{\nu}_{z,t} \geq \sigma^2$, set $\hat{\boldsymbol{\beta}}_t = 0$ or else compute $\hat{\boldsymbol{\beta}}_t$ using Lemma 1
8:      $\forall s \leq t \in \mathbb{N}$ : compute $z_{s,q}$ using $\hat{\boldsymbol{\beta}}_t$ in Eq. (2)
9:      Set $\overline{V}_{t+1} = \overline{V}_t + x_t x_t^\top$, $\hat{\theta}_{t+1}^z = \overline{V}_{t+1}^{-1} \sum_{s=1}^t x_s z_{s,q}$
10: **end for**

---

After selecting an action $x_t$, the algorithm observes the reward $y_t$ with its associated auxiliary feedback $\{w_{t,i}\}_{i=1}^q$. It computes the upper bound on sample variance of hybrid reward (denoted by $\bar{\nu}_{z,t}$[1]) if $t > q + 2$ or else it is set to $\sigma^2$ and then checks two conditions. The first condition (i.e., $t \leq q + 2$) guarantees the sample variance is well-defined. Whereas the second condition (i.e., $\bar{\nu}_{z,t} > \sigma^2$) ensures the algorithm at least be as good as OFUL because $\bar{\nu}_{z,t}$ can be larger than $\sigma^2$ due to the overestimation in initial rounds. If both conditions $t \leq q + 2$ and $\bar{\nu}_{z,t} \geq \sigma^2$ fail, the value of $\hat{\boldsymbol{\beta}}_t$ is re-computed as defined in Lemma 1. The updated $\hat{\boldsymbol{\beta}}_t$ is then used to update all hybrid rewards, i.e., $z_{s,q}$, $\forall s \leq t \in \mathbb{N}$. Finally, the values of $\overline{V}_{t+1}$ and $\hat{\theta}_{t+1}$ are updated as $\overline{V}_{t+1} = \overline{V}_t + x_t x_t^\top$ and $\hat{\theta}_{t+1}^z = \overline{V}_{t+1}^{-1} \sum_{s=1}^t x_s z_{s,q}$, which are then used to select the action in the following round. When $\hat{\boldsymbol{\beta}}_t = 0$ for all hybrid rewards, hybrid rewards are the same as the observed rewards, and hence OFUL-AF is the same as OFUL.

The regret analysis of any bandit algorithm hinges on bounding the instantaneous regret for each action. The following result gives an upper bound on the instantaneous regret of OFUL-AF.

**Theorem 2.** *With a probability of at least $1 - 2\delta$, the instantaneous regret of OFUL-AF in round $t$ is*

$$ r_t(\text{OFUL-AF}) \leq 2 \left( \alpha_t^\sigma + \lambda^{1/2} S \right) \|x_t\|_{\overline{V}_t^{-1}}, $$

*where $\alpha_t^\sigma = \sqrt{\min\left(\sigma^2, \bar{\nu}_{z,t-1}\right)} \, \alpha_t$, $\|\theta^\star\|_2 \leq S$, and $\alpha_t = \sqrt{d \log\left(\frac{1 + t L^2/\lambda}{\delta}\right)}$. For $t > q + 2$ and $\bar{\nu}_{z,t} < \sigma^2$, $\mathbb{E}\left[r_t(\text{OFUL-AF})\right] \leq \widetilde{O}\left( \left(\frac{(t-3)(1-\rho^2)}{t-q-3}\right)^{\frac{1}{2}} r_t(\text{OFUL}) \right)$. Here, $\widetilde{O}$ hides constant terms.*

---

[1]Let $\hat{\nu}_{z,t}$ be the sample variance estimate of hybrid rewards (details in Appendix A.2). Then, $\bar{\nu}_{z,t} = \frac{(t-2)\hat{\nu}_{z,t}}{\chi^2_{1-\delta,t}}$, where $\chi^2_{1-\delta,t}$ denotes $100(1-\delta)^{\text{th}}$ percentile value of the chi-squared distribution with $t - 2$ degrees of freedom.

The proof follows by bounding the estimation error of the parameter $\theta^\star$ when the estimation method uses auxiliary feedback. This result shows that auxiliary feedback leads to a better instantaneous regret upper bound and a better regret (as defined in Eq. (1)) than the vanilla OFUL algorithm. Since the improvement in instantaneous regret increase with $t$, having a single constant to compare with regret of OFUL may lead to weaker regret upper bound than the sum of all instantaneous regret.

## 4 Estimated auxiliary feedback functions

Auxiliary feedback functions may be unknown in many real-life problems. However, the learner can construct an unbiased estimator for the auxiliary feedback function using historical data or acquiring more samples of auxiliary feedback. But these estimated functions offer a lower variance reduction than known auxiliary functions. To study the effect of using the estimated auxiliary feedback functions on the performance of bandit algorithms, we borrow some techniques from approximate control variate theory (Gorodetsky et al., 2020; Pham and Gorodetsky, 2022) as we discussed next.

### 4.1 Approximate control variates

Let $y$ be an unbiased estimator of an unknown variable $\mu$ and a random variable $w$ with a known estimated mean ($\omega_e$) be a control variate of $y$. As long as the known estimated mean is an unbiased estimator of $w$, one can use it to reduce the variance of $y$ as follows. For any choice of a coefficient $\beta_e$, define a new random variable as $z_e \doteq y - \beta_e \bar{w}$, where $\bar{w} = w - \omega_e$. Since $\omega_e$ is an unbiased estimator of $w$, it is straightforward to show that $z_e$ is also an unbiased estimator of $y$.

By using properties of variance and covariance, the variance of $z_e$ is given by

$$\text{Var}(z_e) = \text{Var}(y) + \beta_e^2 \text{Cov}(\bar{w}, \bar{w}) - 2\beta_e \text{Cov}(y, \bar{w}).$$

The variance of $z_e$ is minimized by setting $\beta_e$ to $\beta_e^\star = \text{Cov}(\bar{w}, \bar{w})^{-1}\text{Cov}(y, \bar{w})$ and the minimum value of $\text{Var}(z_e)$ is $(1 - \rho_e^2)\text{Var}(y)$, where $\rho_e = \text{Cov}(y, \bar{w})\left(\text{Cov}(\bar{w}, \bar{w})^{-1}/\text{Var}(y)\right)\text{Cov}(y, \bar{w})$.

### 4.2 Auxiliary feedback with unknown functions as approximate control variates

We now introduce a new definition of *hybrid reward* that uses estimated auxiliary feedback functions. Let $w_{s,i}$ be the $i^{\text{th}}$ auxiliary feedback observed with reward $y_s$ and $g_{e,i}$ be the unbiased estimator of function $g_i$. Then, the hybrid reward with $q$ estimated auxiliary feedback functions is defined by

$$z_{e,s,q} = y_s - \sum_{i=1}^{q} \beta_{e,i}(w_{s,i} - g_{e,i}(x_t)) = y_s - (\boldsymbol{w}_s - \boldsymbol{g}_{e,s})\boldsymbol{\beta}_e. \tag{3}$$

where $\boldsymbol{w}_s = (w_{s,1}, \ldots, w_{s,q})$, $\boldsymbol{g}_{e,s} = (g_{e,1}(x_s), \ldots, g_{e,q}(x_s))$, and $\boldsymbol{\beta}_e = (\beta_{e,1}, \ldots, \beta_{e,q})^\top$. Let $\Sigma_{\bar{\boldsymbol{w}}\bar{\boldsymbol{w}}} \in \mathbb{R}^{q \times q}$ denote the covariance matrix among centered auxiliary feedback (i.e., $\bar{\boldsymbol{w}}_s = \boldsymbol{w}_s - \boldsymbol{g}_{e,s}$), and $\sigma_{\boldsymbol{y}\bar{\boldsymbol{w}}} \in \mathbb{R}^{q \times 1}$ denote the vector of covariance between reward and its centered auxiliary feedback. Then, the variance of $z_{e,s,q}$ is minimized by setting the $\boldsymbol{\beta}_e$ to $\boldsymbol{\beta}_e^\star = \Sigma_{\bar{\boldsymbol{w}}\bar{\boldsymbol{w}}}^{-1}\sigma_{\boldsymbol{y}\bar{\boldsymbol{w}}}$, and the minimum value of $\text{Var}(z_{e,s,q})$ is $(1 - \rho_e^2)\sigma^2$, where $\rho_e^2 = \sigma_{\boldsymbol{y}\bar{\boldsymbol{w}}}^\top \Sigma_{\bar{\boldsymbol{w}}\bar{\boldsymbol{w}}}^{-1} \sigma_{\boldsymbol{y}\bar{\boldsymbol{w}}}/\sigma^2$.

The definition of hybrid reward given in Eq. (3) is very flexible and allows different estimators to estimate auxiliary feedback functions. The only difference among these estimators is how they partition the available samples of auxiliary feedback to estimate auxiliary function $g_i$. As no optimal partitioning strategy is known, we adopt the Independent Samples (IS) and Multi-Fidelity (MF) sampling strategy for our setting where finite samples of auxiliary feedback are available. Both strategies are proven to be asymptotically optimal (Gorodetsky et al., 2020), implying the variance reduction is asymptotically the same as if auxiliary feedback functions are known.

**IS and MF sampling strategy:** Let $s$ and $s_i \supset s$ be the sample sets used for estimating functions $f$ and $g_i$, respectively. Then, for the IS sampling strategy, $(s_i \setminus s) \cap (s_j \setminus s) = \varnothing$ for $i \neq j$, i.e., the extra samples used for estimating the function $g_i$ are unique. Whereas, for the MF sampling strategy, $s_i = s \cup_{j=1}^{i} s_j'$ and $s_i' \cap s_j' = \varnothing$ for $i \neq j$, i.e., the estimation of function $g_i$ uses the samples that were used for estimating function $g_{i-1}$ with some additional samples. Refer to Fig. 1 for the visual representation of both sampling strategies. After adopting Theorem 3 and Theorem 4 from

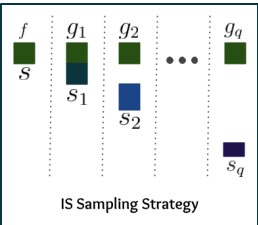 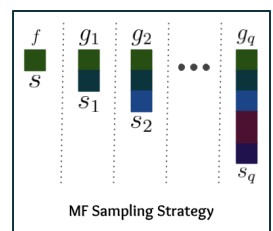 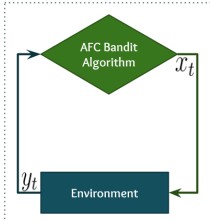 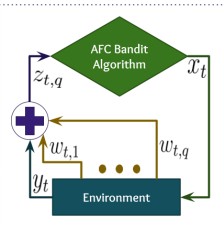

Figure 1: **Left two figures:** Visualization of IS and MF sampling strategies. Each column represents samples used for estimating function (written at the top), and the same color is used to show shared samples among auxiliary function estimation. **Right two figures:** Interaction between AFC bandit algorithm and environment. AFC bandit algorithm that only uses observed rewards (second from right), and AFC bandit algorithm that also uses auxiliary feedback as hybrid rewards (rightmost).

Gorodetsky et al. (2020) to our setting, we can further simplify $\Sigma_{\bar{\boldsymbol{w}}\bar{\boldsymbol{w}}}$ and $\sigma_{\boldsymbol{y}\bar{\boldsymbol{w}}}$ when IS and MF sampling strategies (denoted by $e$) are used for estimating auxiliary feedback functions as follows:

$$\Sigma_{\bar{\boldsymbol{w}}\bar{\boldsymbol{w}}} = (\Sigma_{\boldsymbol{w}\boldsymbol{w}} \circ \boldsymbol{F}_e)/t \text{ and } \sigma_{\boldsymbol{y}\bar{\boldsymbol{w}}} = (\text{diag}\,(\boldsymbol{F}_e) \circ \sigma_{\boldsymbol{y}\boldsymbol{w}})/t,$$

where $t$ denotes the number of reward observations with its auxiliary feedback, $\text{diag}(A)$ represents a vector whose elements are the diagonal of the matrix $A$, and $\circ$ denotes an element-wise product. The $ij$-th element of matrix $\boldsymbol{F}_e \in \mathbb{R}^{q \times q}$ is

$$f_{e,ij} = \begin{cases} ((r_i - 1)(r_j - 1))/(r_i r_j) & \text{if } i \neq j \text{ and } e = \text{IS} \\ (\min(r_i, r_j) - 1)/\min(r_i, r_j) & \text{if } i \neq j \text{ and } e = \text{MF} \\ (r_i - 1)/r_i & \text{otherwise,} \end{cases}$$

where $r_i \in \mathbb{R}^+$ is the ratio between the total number of samples used for estimating function $g_i$ by sampling strategy $e$ and the total number of samples used for estimating $f$.

Since $\Sigma_{\bar{\boldsymbol{w}}\bar{\boldsymbol{w}}}$ and $\sigma_{\boldsymbol{y}\bar{\boldsymbol{w}}}$ may be unknown, they must be estimated to get the best estimate for $\boldsymbol{\beta}^\star$. Our following result gives the best linear unbiased estimator of $\boldsymbol{\beta}_e$ (i.e., $\hat{\boldsymbol{\beta}}_{e,t}$) that uses $t$ observations of rewards and their auxiliary feedback with estimated auxiliary feedback functions.

**Lemma 2.** *Let $t > q + 2 \in \mathbb{N}$, $e$ is the sampling strategy, and $f_t$ be the estimate of function $f$ at the end of round $t$ which uses $\{x_s, y_s, \boldsymbol{w}_s\}_{s=1}^t$. Then, the best linear unbiased estimator of $\boldsymbol{\beta}_e^\star$ is*

$$\hat{\boldsymbol{\beta}}_{e,t} = (\boldsymbol{W}_t^\top \boldsymbol{W}_t \circ \boldsymbol{F}_e)^{-1} \left( diag\,(\boldsymbol{F}_e) \circ \boldsymbol{W}_t^\top \boldsymbol{Y}_t \right),$$

*where $\boldsymbol{W}_t$ is a $t \times q$ matrix whose $s^{th}$ row is $\boldsymbol{w}_s - \boldsymbol{g}_{e,s}$ and $\boldsymbol{Y}_t = (y_1 - f_t(x_1), \ldots, y_t - f_t(x_t))$.*

After adopting matrix manipulation tricks from Pham and Gorodetsky (2022) to our setting, the proof follows similar steps as the proof of Lemma 1. We now characterize the properties of the hybrid reward that uses either IS or MF sampling strategy for estimating auxiliary feedback functions.

**Theorem 3.** *Let $t > q + 2 \in \mathbb{N}$ and $e$ is the sampling strategy. If $\hat{\boldsymbol{\beta}}_{e,t}$ as defined in Lemma 2 is used to compute hybrid reward $z_{e,s,q}$ for any $s \leq t \in \mathbb{N}$, then $\mathbb{E}\,[z_{e,s,q}] = f(x_s)$ and $\mathbb{V}\,(z_{e,s,q}) = \left(1 + \frac{a(e)q}{t-q-2}\right)(1 - \rho_e^2)\sigma^2$, where $a(IS) = 1$, $a(MF) = \frac{r-1}{r}$ if $r_i = r$, $\forall i \in \{1, 2, \ldots, q\}$ when using MF sampling strategy for estimating auxiliary feedback functions.*

The key takeaways from Theorem 3 are as follows. First, the hybrid reward with estimated auxiliary feedback is still an unbiased estimator, so one can use it to estimate the reward function $f$. Second, there is a potential loss in variance reduction as it has an extra multiplicative factor $a(e)$ and $\rho_e^2 \leq \rho^2$.

*Remark* 1. As samples for estimating auxiliary functions increase compared to the reward function, the variance reduction from IS and MF sampling strategy converges to the reduction achieved using known auxiliary functions. As $\forall i : r_i \to \infty$, then $\boldsymbol{F}_e \to \boldsymbol{1}_{q \times q}$. It is now straightforward to see that $\Sigma_{\bar{\boldsymbol{w}}\bar{\boldsymbol{w}}}$ will become $\Sigma_{\boldsymbol{w}\boldsymbol{w}}$, $\sigma_{\boldsymbol{y}\bar{\boldsymbol{w}}}$ will become $\sigma_{\boldsymbol{y}\boldsymbol{w}}$, and hence $\rho_e^2 = \rho^2$ as $\forall i : r_i \to \infty$.

*Remark* 2. The IS and MF sampling strategies are shown to be asymptotically optimal (Gorodetsky et al., 2020), i.e., the variance reduction achieved by both strategies is asymptotically the same as if

auxiliary feedback functions are known. However, both sampling strategies are useful in different applications, e.g., the IS sampling strategy suits the problems in which different auxiliary feedback can be independently sampled. In contrast, the MF sampling suits the problems where auxiliary feedback can not be sampled independently.

### 4.3 Parameterized bandits with estimated auxiliary feedback functions

We now consider the parameterized bandit setting described in Section 2, where the reward and auxiliary feedback function can be non-linear. To exploit the available auxiliary feedback in linear bandits, we propose a method in Section 3.3 that uses hybrid reward in place of rewards to get tight upper confidence bound for the estimator of an unknown reward function and hence smaller regret as compared to the vanilla OFUL due to the smaller variance of the hybrid rewards. We generalize this observation and introduce the notion of *Auxiliary Feedback Compatible (AFC)* bandit algorithm.

**Definition 1** (**AFC Bandit Algorithm**). Any bandit algorithm $\mathfrak{A}$ is *Auxiliary Feedback Compatible* if: (i) $\mathfrak{A}$ can use correlated reward samples to construct upper confidence bound for reward function and (ii) with probability $1 - \delta$, its estimated reward function $f_t^{\mathfrak{A}}$ has the following property:

$$|f_t^{\mathfrak{A}}(x) - f(x)| \leq \sigma h(x, \mathcal{O}_t) + l(x, \mathcal{O}_t),$$

where $x \in \mathcal{X}$, $\sigma^2$ is the variance of Gaussian noise in observed reward, and $\mathcal{O}_t$ denotes the observations of actions and their rewards with the parameters of $\mathfrak{A}$ at the beginning of round $t$.

As the estimated coefficient vector uses all past samples, the resultant hybrid rewards become correlated due to using this estimated coefficient vector. Bandit algorithms like UCB1 (Auer et al., 2002) and kl-UCB (Cappé et al., 2013) are not AFC as they need independent reward samples to construct upper confidence bounds. In contrast, bandit algorithms like OFUL (Abbasi-Yadkori et al., 2011), Lin-UCB (Chu et al., 2011), UCB-GLM (Li et al., 2017), IGP-UCB, and GP-TS (Chowdhury and Gopalan, 2017) are AFC as they all use techniques proposed in Abbasi-Yadkori et al. (2011) for building the upper confidence bound, which does not need reward samples to be independent.

As AFC bandit algorithms use the noise variance of observed reward for constructing the confidence upper bound, they can also exploit available auxiliary feedback by replacing reward with its respective hybrid reward as shown in Fig. 1 (rightmost figure). We next give an upper bound on the instantaneous regret for any AFC bandit algorithm that uses hybrid rewards instead of observed rewards.

**Theorem 4.** *Let $\mathfrak{A}$ be an AFC bandit algorithm with $|f_t^{\mathfrak{A}}(x) - f(x)| \leq \sigma h(x, \mathcal{O}_t) + l(x, \mathcal{O}_t)$ and $\bar{\nu}_{e,z,t}$ be the upper bound on sample variance of hybrid reward, whose value is set to $\sigma^2$ for $t \leq q + 2$. Then, with a probability of at least $1 - 2\delta$, the instantaneous regret of $\mathfrak{A}$ after using hybrid rewards (named $\mathfrak{A}$-AF) for reward function estimation in round $t$ is*

$$r_t(\mathfrak{A}\text{-AF}) \leq 2\min(\sigma, (\bar{\nu}_{e,z,t})^{\frac{1}{2}})h(x, \mathcal{O}_t) + l(x, \mathcal{O}_t),$$

*where $e = \{IS, MF, KF\}$, and KF denotes the case where auxiliary functions are known. For $t > q+2$ and $\bar{\nu}_{e,z,t} < \sigma^2$, $\mathbb{E}\left[r_t(\mathfrak{A}\text{-AF})\right] \leq \widetilde{O}\left(\left(\left(\frac{t-(1-a(e))q-3}{t-q-3}\right)(1-\rho_e^2)\right)^{\frac{1}{2}} r_t(\mathfrak{A})\right)$, where $a(KF) = 1$.*

After using Theorem 1 and Theorem 3 to replace the variance of hybrid reward, the proof follows similar steps as the proof of Theorem 2. We have given more details about the values of $h(x, \mathcal{O}_t)$ and $l(x, \mathcal{O}_t)$ for different AFC bandit algorithms in Table 1.

Table 1: Values of $h(x, \mathcal{O}_t)$ and $l(x, \mathcal{O}_t)$ for different AFC bandit algorithms

| AFC bandit algorithm | $h(x, \mathcal{O}_t)$ | $l(x, \mathcal{O}_t)$ |
|---|---|---|
| OFUL (Abbasi-Yadkori et al., 2011) | $\sqrt{d\log\left(\frac{1+tL^2/\lambda}{\delta}\right)}\|x\|_{\overline{V}_t^{-1}}$ | $\lambda^{\frac{1}{2}}S\|x\|_{\overline{V}_t^{-1}}$ |
| Lin-UCB (OFUL for contextual setting) | $\sqrt{d\log\left(\frac{1+tL^2/\lambda}{\delta}\right)}\|x\|_{\overline{V}_t^{-1}}$ | $\lambda^{\frac{1}{2}}S\|x\|_{\overline{V}_t^{-1}}$ |
| GLM-UCB (Li et al., 2017) | $\sqrt{\frac{d}{2}\log(1+2t/d)+\log(1/\delta)}\frac{\|x\|_{V_t^{-1}}}{\kappa}$ | $0$ |
| IGP-UCB (Chowdhury and Gopalan, 2017) | $\sqrt{2(\gamma_{t-1}+1+\log(1/\delta))}\sigma_{t-1}(x)$ | $B\sigma_{t-1}(x)$ |

# 5   Experiments

To validate our theoretical results, we empirically demonstrate the performance gain due to auxiliary feedback in different settings of parameterized bandits. We repeat all our experiments 50 times and show the regret as defined in Eq. (1) with a $95\%$ confidence interval (vertical line on each curve shows the confidence interval). Due to space constraints, the details of used problem instances are given in Appendix A.5 of the supplementary material.

**Comparing regret with benchmark bandit algorithms:**    We considered three bandit settings for our experiments: linear bandits, linear contextual bandits, and non-linear contextual bandits (details are given in Appendix A.5). The formal setting of a contextual bandits with auxiliary feedback is given in Appendix A.4. We used the following existing bandit algorithms for these settings: OFUL (Abbasi-Yadkori et al., 2011) for linear bandits, Lin-UCB (Chu et al., 2011) for linear contextual bandits, and Lin-UCB with the polynomial kernel (which we named NLin-UCB) for non-linear contextual bandits. We compare the performance of these benchmark bandit algorithms with four different variants of our algorithms. The first variant assumes the auxiliary feedback functions are known (highlighted by adding '-AF' to the benchmark algorithms). When auxiliary feedback functions are unknown, we use IS and MF sampling strategy while maintaining $r = 2$ (i.e., getting one extra sample of auxiliary feedback in each round). The IS and MF sampling strategies are the same when only one auxiliary feedback exists. Since we only use one auxiliary feedback in our experiments, we highlight this variant by adding '-IS/MF' to the benchmark algorithms. When IS and MF sampling strategies are used, one needs to update the auxiliary feedback functions in each round to get better estimators. However, it leads to the re-computation of all variables that are needed for updating the hybrid rewards, which is not needed when auxiliary feedback functions are fixed. Therefore, we consider two more computationally efficient variants for the unknown auxiliary functions setting. One variant assumes the knowledge of biased auxiliary feedback, i.e., $g_i(x) + \varepsilon_g$ is available instead of $g_i(x)$ (highlighted by adding '-BE' to the benchmark algorithms). Another variant assumes that some initial samples of auxiliary feedback are available, which are used to get the auxiliary feedback function estimator. We highlight this variant by adding '-EH' to the benchmark algorithms. All variants with given parameters perform better than benchmark bandit algorithms (see Fig. 2a, Fig. 2b, and Fig. 2c). We observe the expected performance among these variants as the variant with a known auxiliary feedback function outperforms all other variants. At the same time, IS/MF sampling strategy-based variant outperforms the other two heuristic variants for the setting of unknown auxiliary feedback function.

**Regret vs. different biased estimator:** To know the effect of bias in auxiliary feedback (i.e., $\varepsilon_g$) in the mean value of auxiliary feedback, we run an experiment with the same linear contextual bandits experiment setup mentioned above. To see the variation in regret, we set $\varepsilon_g = \{1, 0.2, 0.1, 0.07, 0.05\}$. As shown in Fig. 2d, the regret increases with an increase in bias and even starts performing poorly than Lin-UCB. This experiment demonstrates that as long as the bias in auxiliary feedback is within a limit, there will be an advantage to using this computationally efficient variant.

**Regret vs. number of historical samples of auxiliary feedback :** Increasing the number of historical samples of auxiliary feedback for estimating the auxiliary feedback function reduces the error in its estimation, leading to better performance. To observe this, we use estimators using different numbers of auxiliary feedback samples, i.e., $n_h = \{5, 7, 10, 15, 20\}$ in linear contextual bandits setting. As expected, the regret decreases with an increase in auxiliary feedback samples, but using an estimator with a few samples even performs poorly than Lin-UCB, as shown in Fig. 2e.

**Regret vs. correlation coefficient:** As theoretical results imply that the regret decreases when the correlation between reward and auxiliary feedback increases. To validate this, we used problem instances with different correlation coefficients in linear contextual bandits setting. As expected, we observe that the regret decreases as the correlation coefficient increases, as shown in Fig. 2f.

# 6   Conclusion

This paper studies a novel parameterized bandit problem in which a learner observes auxiliary feedback correlated with the observed reward. We first introduce the notion of 'hybrid reward,' which combines the reward and its auxiliary feedback. To get the maximum benefit from hybrid reward, we treat auxiliary feedback as a control variate and then extend control variate theory to a setting

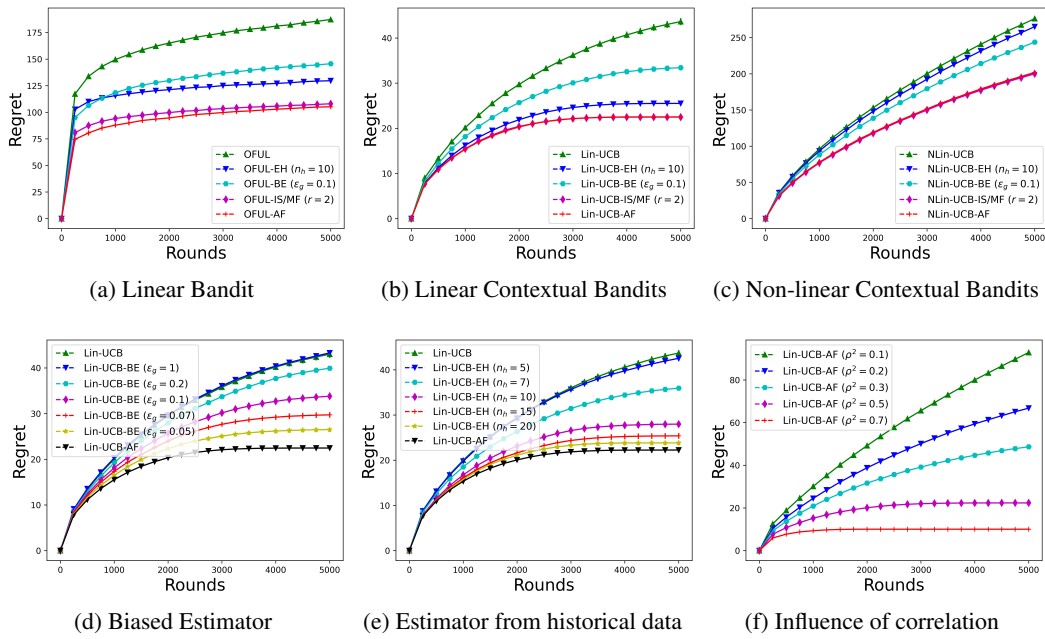

| | | |
|---|---|---|
| (a) Linear Bandit | (b) Linear Contextual Bandits | (c) Non-linear Contextual Bandits |
| (d) Biased Estimator | (e) Estimator from historical data | (f) Influence of correlation |

Figure 2: **Top row:** Comparing regret of different variants with their benchmark bandit algorithms in different settings. **Bottom row:** Regret vs. different biases in Lin-UCB-BE (left figure), regret vs. number of historical samples of auxiliary feedback in Lin-UCB-EH (middle figure), and regret of Lin-UCB-AF vs. varying correlation coefficients of reward and its auxiliary feedback (right figure).

where a function can parameterize control variates. Equipped with these results, we show that the variance of hybrid rewards is smaller than observed rewards. We then use these hybrid rewards to estimate the reward function, leading to tight confidence bounds and hence smaller regret. We have proved that the expected instantaneous regret of any AFC bandit algorithm after using hybrid rewards is improved by a factor of $O((1 - \rho^2)^{\frac{1}{2}})$, where $\rho$ is the correlation coefficient of the reward and its auxiliary feedback. Our experiments also validate our theoretical results. An interesting future direction is to extend these results to bandit settings with heteroscedastic and non-Gaussian noise.

## Acknowledgments and Disclosure of Funding

DesCartes: this research is supported by the National Research Foundation, Prime Minister's Office, Singapore under its Campus for Research Excellence and Technological Enterprise (CREATE) programme.

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

# A  Supplementary material

## A.1  Missing proofs related to auxiliary feedback

**Results from linear regression**

We first state results for linear regression that we will use in the subsequent proofs. Consider the following regression problem with $t$ samples and $q$ features:

$$z_s = \boldsymbol{x}_s^\top \boldsymbol{\theta} + \varepsilon_s, \quad i \in \{1, 2, \ldots, t\}$$

where $z_s \in \mathbb{R}$ is the $s^{\text{th}}$ target variable, $\boldsymbol{x}_s = (x_{s1}, \ldots, x_{sq}) \in \mathbb{R}^q$ is the $s^{\text{th}}$ feature vector, $\boldsymbol{\theta} \in \mathbb{R}^q$ is the unknown regression parameters, and $\varepsilon_s$ is a normally distributed noise with mean 0 and constant variance $\sigma^2$. The values of noise $\varepsilon_s$ form an IID sequence and are independent of $\boldsymbol{x}_s$. Let

$$\boldsymbol{Z}_t = \begin{pmatrix} z_1 \\ \vdots \\ z_t \end{pmatrix}, \qquad \boldsymbol{X}_t = \begin{pmatrix} x_{11} & \cdots & x_{1q} \\ \vdots & \cdots & \vdots \\ X_{t1} & \cdots & x_{tq} \end{pmatrix}, \quad \text{and} \quad \boldsymbol{\varepsilon}_t = \begin{pmatrix} \varepsilon_1 \\ \vdots \\ \varepsilon_t \end{pmatrix}.$$

Then, the best linear unbiased estimator of $\boldsymbol{\theta}$ is $\hat{\boldsymbol{\theta}}_t = (\boldsymbol{X}_t^\top \boldsymbol{X}_t)^{-1} \boldsymbol{X}_t^\top \boldsymbol{Z}_t$, which has the following finite sample properties.

*Fact* 1. The following are the finite sample properties of the least square estimator $\hat{\boldsymbol{\theta}}_t$:

$\quad$ 1. $\mathbb{E}\left[\hat{\boldsymbol{\theta}}_t | \boldsymbol{X}_t\right] = \boldsymbol{\theta}$, $\hspace{4cm}$ (unbiased estimator)

$\quad$ 2. $\text{Var}(\hat{\boldsymbol{\theta}}_t | \boldsymbol{X}_t) = \sigma^2 (\boldsymbol{X}_t^\top \boldsymbol{X}_t)^{-1}$, and $\hspace{1.5cm}$ (expression for the variance)

$\quad$ 3. $\text{Var}(\hat{\boldsymbol{\theta}}_{ti} | \boldsymbol{X}_t) = \sigma^2 (\boldsymbol{X}_t^\top \boldsymbol{X}_t)_{ii}^{-1}$, $\hspace{2cm}$ (element-wise variance)

where $(\boldsymbol{X}_t^\top \boldsymbol{X})_{ii}^{-1}$ is the $ii-$element of the matrix $(\boldsymbol{X}_t^\top \boldsymbol{X})^{-1}$.

In the above result, the first two properties are from Proposition 1.1 of Hayashi (2000), whereas the third property is from Van De Geer (2005). The following result gives the finite sample properties of the estimator of noise variance $\sigma^2$.

*Fact* 2. (Hayashi, 2000, Proposition 1.2) Let $\hat{\sigma}_t^2 = \frac{1}{t-q} \sum_{s=1}^t (z_s - \boldsymbol{x}_s^\top \hat{\boldsymbol{\theta}}_t)^2$ be estimator of $\sigma^2$ and $t > q$ (so that $\hat{\sigma}_t^2$ is well defined). Then, $\hat{\sigma}_t^2$ is an unbiased estimator of $\sigma^2$, i.e., $\mathbb{E}\left[\hat{\sigma}_t^2 | \boldsymbol{X}_t\right] = \sigma^2$.

Using the Schur complement, we have the following results about the inverse of the block matrix.

*Fact* 3. Let $G = \begin{pmatrix} t & B \\ C & D \end{pmatrix}$ be a block matrix, where $t \in \mathbb{R} \setminus \{0\}$, $B$, $C$, $D$ are respectively $1 \times q$, $q \times 1$, and $q \times q$ matrices of real numbers. Then, $G_{11}^{-1} = t^{-1} + t^{-1} B (tD - CB)^{-1} C$.

**Control variates theory**

Let $y$ be the random variable of interest with unknown mean $\mu$. There are $q$ control variates correlated with $y$, where $i^{\text{th}}$ control variate has mean $\omega_i$ and its $s^{\text{th}}$ observation is denoted by $w_{s,i}$. For any $s \in \{1, \ldots, t\}$, we define a variable $z_s$ using $s^{\text{th}}$ observation of $y_s$ and its control variates as follows:

$$z_s = y_s - (\boldsymbol{w}_s - \boldsymbol{\omega})\boldsymbol{\beta},$$

where $\boldsymbol{w}_s = (w_{s,1}, \ldots, w_{s,q})$ and $\boldsymbol{\omega} = (\omega_1, \ldots, \omega_q)$. The above equation can be re-written as:

$$y_s = z_s + (\boldsymbol{w}_s - \boldsymbol{\omega})\boldsymbol{\beta}.$$

Under the assumption of $z_s$ being a unbiased estimator of $\mu$, we can write $y_s$ as follows:

$$y_s = \mu + (\boldsymbol{w}_s - \boldsymbol{\omega})\boldsymbol{\beta} + \varepsilon_{z,s}.$$

where $\varepsilon_{z,1}, \ldots, \varepsilon_{z,t}$ are IID and have zero mean Gaussian noise with variance $(1 - \rho^2)\sigma^2$. Let

$$\overline{\boldsymbol{Y}}_t = \begin{pmatrix} y_1 \\ \vdots \\ y_t \end{pmatrix}, \; \overline{\boldsymbol{W}}_t = \begin{pmatrix} 1 & \boldsymbol{w}_1 - \boldsymbol{\omega} \\ \vdots & \vdots \\ 1 & \boldsymbol{w}_t - \boldsymbol{\omega} \end{pmatrix}, \; \boldsymbol{\gamma} = \begin{pmatrix} \mu \\ \boldsymbol{\beta} \end{pmatrix}, \; \text{and } \boldsymbol{\varepsilon}_{z,t} = \begin{pmatrix} \varepsilon_{z,1} \\ \vdots \\ \varepsilon_{z,t} \end{pmatrix}.$$

The best linear unbiased estimator of $\boldsymbol{\gamma}$ is $\hat{\boldsymbol{\gamma}} = (\overline{\boldsymbol{W}}_t^\top \overline{\boldsymbol{W}}_t)^{-1} \overline{\boldsymbol{W}}_t^\top \overline{\boldsymbol{Y}}_t$. To get $\hat{\mu}_{z,t}$ and $\hat{\beta}^\star$, we expand $\hat{\boldsymbol{\gamma}}$ as follows:

$$\hat{\boldsymbol{\gamma}} = \left( \begin{pmatrix} 1 & \boldsymbol{w}_1 - \boldsymbol{\omega} \\ \vdots & \vdots \\ 1 & \boldsymbol{w}_t - \boldsymbol{\omega} \end{pmatrix}^\top \begin{pmatrix} 1 & \boldsymbol{w}_1 - \boldsymbol{\omega} \\ \vdots & \vdots \\ 1 & \boldsymbol{w}_t - \boldsymbol{\omega} \end{pmatrix} \right)^{-1} \begin{pmatrix} 1 & \boldsymbol{w}_1 - \boldsymbol{\omega} \\ \vdots & \vdots \\ 1 & \boldsymbol{w}_t - \boldsymbol{\omega} \end{pmatrix}^\top \begin{pmatrix} y_1 \\ \vdots \\ y_t \end{pmatrix}$$

$$= \left( \begin{pmatrix} 1 & \cdots & 1 \\ \boldsymbol{w}_1 - \boldsymbol{\omega} & \cdots & \boldsymbol{w}_t - \boldsymbol{\omega} \end{pmatrix} \begin{pmatrix} 1 & \boldsymbol{w}_1 - \boldsymbol{\omega} \\ \vdots & \vdots \\ 1 & \boldsymbol{w}_t - \boldsymbol{\omega} \end{pmatrix} \right)^{-1} \begin{pmatrix} 1 & \cdots & 1 \\ \boldsymbol{w}_1 - \boldsymbol{\omega} & \cdots & \boldsymbol{w}_t - \boldsymbol{\omega} \end{pmatrix} \begin{pmatrix} y_1 \\ \vdots \\ y_t \end{pmatrix}$$

$$= \begin{pmatrix} t & \sum_{s=1}^t (\boldsymbol{w}_s - \boldsymbol{\omega}) \\ \sum_{s=1}^t (\boldsymbol{w}_s - \boldsymbol{\omega}) & \sum_{s=1}^t (\boldsymbol{w}_s - \boldsymbol{\omega})^\top (\boldsymbol{w}_s - \boldsymbol{\omega}) \end{pmatrix}^{-1} \begin{pmatrix} \sum_{s=1}^t y_s \\ \sum_{s=1}^t (\boldsymbol{w}_s - \boldsymbol{\omega}) y_s \end{pmatrix}$$

After taking first matrix from RHS to LHS and using $\hat{\boldsymbol{\gamma}} = \begin{pmatrix} \hat{\mu}_{z,t} \\ \hat{\boldsymbol{\beta}}_t \end{pmatrix}$, we have

$$\begin{pmatrix} t & \sum_{s=1}^t (\boldsymbol{w}_s - \boldsymbol{\omega}) \\ \sum_{s=1}^t (\boldsymbol{w}_s - \boldsymbol{\omega}) & \sum_{s=1}^t (\boldsymbol{w}_s - \boldsymbol{\omega})^\top (\boldsymbol{w}_s - \boldsymbol{\omega}) \end{pmatrix} \begin{pmatrix} \hat{\mu}_{z,t} \\ \hat{\boldsymbol{\beta}}_t \end{pmatrix} = \begin{pmatrix} \sum_{s=1}^t y_s \\ \sum_{s=1}^t (\boldsymbol{w}_s - \boldsymbol{\omega}) y_s \end{pmatrix}. \tag{4}$$

From above, we get the following equation:

$$t\hat{\mu}_{z,t} + \left( \sum_{s=1}^t (\boldsymbol{w}_s - \boldsymbol{\omega}) \right) \hat{\boldsymbol{\beta}}_t = \sum_{s=1}^t y_s$$

$$\implies \hat{\mu}_{z,t} = \frac{1}{t} \sum_{s=1}^t y_s - \left( \frac{1}{t} \sum_{s=1}^t (\boldsymbol{w}_s - \boldsymbol{\omega}) \right) \hat{\boldsymbol{\beta}}_t.$$

Using $\hat{\mu}_{y,t} = \frac{1}{t} \sum_{s=1}^t y_s$ and $\hat{\boldsymbol{\omega}}_t = \frac{1}{t} \sum_{s=1}^t \boldsymbol{w}_s$, we get

$$\hat{\mu}_{z,t} = \hat{\mu}_{y,t} - (\hat{\boldsymbol{\omega}}_t - \boldsymbol{\omega}) \hat{\boldsymbol{\beta}}_t. \tag{5}$$

Similarly, we have another equation as follows:

$$\hat{\mu}_{z,t} \sum_{s=1}^t (\boldsymbol{w}_s - \boldsymbol{\omega}) + \left( \sum_{s=1}^t (\boldsymbol{w}_s - \boldsymbol{\omega})^\top (\boldsymbol{w}_s - \boldsymbol{\omega}) \right) \hat{\boldsymbol{\beta}}_t = \sum_{s=1}^t (\boldsymbol{w}_s - \boldsymbol{\omega}) y_s$$

$$\implies \hat{\boldsymbol{\beta}}_t = \left( \sum_{s=1}^t (\boldsymbol{w}_s - \boldsymbol{\omega})^\top (\boldsymbol{w}_s - \boldsymbol{\omega}) \right)^{-1} \left( \sum_{s=1}^t (\boldsymbol{w}_s - \boldsymbol{\omega})(y_s - \hat{\mu}_{z,t}) \right).$$

Using $\boldsymbol{W}_t = \begin{pmatrix} \boldsymbol{w}_1 - \boldsymbol{\omega} \\ \vdots \\ \boldsymbol{w}_t - \boldsymbol{\omega} \end{pmatrix}$ and $\boldsymbol{Y}_t = \begin{pmatrix} y_1 - \hat{\mu}_{z,t} \\ \vdots \\ y_t - \hat{\mu}_{z,t} \end{pmatrix}$, we have

$$\implies \hat{\boldsymbol{\beta}}_t = (\boldsymbol{W}_t^\top \boldsymbol{W}_t)^{-1} \boldsymbol{W}_t^\top \boldsymbol{Y}_t. \tag{6}$$

In the following, we first state the fundamental results from the control variates theory.

*Fact* 4. (Nelson, 1990, Theorem 1) Let $O_s = (Y_s, W_{s,1}, \ldots, W_{s,q})^\top$ follow a $(q+1)-$variate normal distribution with mean vector $(\mu, \boldsymbol{\omega})$ and $\{O_1, \ldots, O_t\}$ be a IID sequence. Assume $\hat{\mu}_{z,t} = \sum_{s=1}^t z_s$, where $z_s = y_s - (\boldsymbol{w}_s - \boldsymbol{\omega})\hat{\boldsymbol{\beta}}_t$ and $\hat{\boldsymbol{\beta}}_t$ used here is given by Eq. (6), then

$$\mathbb{E}[\hat{\mu}_{z,t}] = \mu \text{ and}$$

$$\mathbb{V}(\hat{\mu}_{z,t}) = \left( 1 + \frac{q}{t - q - 2} \right)(1 - \rho^2) \mathbb{V}(\hat{\mu}_{y,t}),$$

where $\sigma_{Y\boldsymbol{W}} \Sigma_{\boldsymbol{W}\boldsymbol{W}}^{-1} \sigma_{Y\boldsymbol{W}}^\top / \sigma^2$ is the square of the multiple correlation coefficient, $\sigma^2 = \mathbb{V}(Y)$, and $\sigma_{Y\boldsymbol{W}} = (\mathrm{Cov}(Y, W_1), \ldots, \mathrm{Cov}(Y, W_q))$ (we have dropped the subscript $s$ as observations are IID).

**Auxiliary feedback as control variates**

**Lemma 1.** *Let $t > q + 2 \in \mathbb{N}$ and $f_t$ be the estimate of function $f$ which uses all information available at the end of round t, i.e., $\{x_s, y_s, \boldsymbol{w}_s\}_{s=1}^t$. Then, the best linear unbiased estimator of $\boldsymbol{\beta}^\star$ is*

$$\hat{\boldsymbol{\beta}}_t \doteq (\boldsymbol{W}_t^\top \boldsymbol{W}_t)^{-1} \boldsymbol{W}_t^\top \boldsymbol{Y}_t,$$

*where $\boldsymbol{W}_t$ is a $t \times q$ matrix whose $s^{th}$ row is $(\boldsymbol{w}_s - \boldsymbol{g}_s)$ and $\boldsymbol{Y}_t = (y_1 - f_t(x_1), \ldots, y_t - f_t(x_t))$.*

*Proof.* Recall Eq. (2) for $s^{th}$ hybrid reward with known auxiliary functions, i.e., $z_{s,q} = y_s - (\boldsymbol{w}_s - \boldsymbol{g}_s)\boldsymbol{\beta}$, which can be re-written as $y_s = z_{s,q} + (\boldsymbol{w}_s - \boldsymbol{g}_s)\boldsymbol{\beta}$. By definition, $z_{s,q} = f(x_s) + \varepsilon_s^z$ for optimal $\beta$, where $\varepsilon_{z,s}$ is zero-mean Gaussian noise with variance $(1 - \rho^2)\sigma^2$. Then, $y_s = f(x_s) + (\boldsymbol{w}_s - \boldsymbol{g}_s)\boldsymbol{\beta} + \varepsilon_{z,s}$. Let $\varphi$ be an unknown function that maps every $x$ to a space where $f(x) = \varphi(x)^\top f$ holds. Then we can re-write the above equation as follows:

$$y_s = f^\top \varphi(x_s) + (\boldsymbol{w}_s - \boldsymbol{g}_s)\boldsymbol{\beta} + \varepsilon_{z,s}.$$

Adapting Eq. (4) to our setting, we have

$$\begin{pmatrix} \sum_{s=1}^t \varphi(x_s)^\top \varphi(x_s) & \sum_{s=1}^t \varphi(x_s)^\top (\boldsymbol{w}_s - \boldsymbol{g}_s) \\ \sum_{s=1}^t (\boldsymbol{w}_s - \boldsymbol{g}_s)^\top \varphi(x_s) & \sum_{s=1}^t (\boldsymbol{w}_s - \boldsymbol{g}_s)^\top (\boldsymbol{w}_s - \boldsymbol{g}_s) \end{pmatrix} \begin{pmatrix} f_t \\ \hat{\boldsymbol{\beta}}_t \end{pmatrix} = \begin{pmatrix} \sum_{s=1}^t \varphi(x_s) y_s \\ \sum_{s=1}^t (\boldsymbol{w}_s - \boldsymbol{g}_s) y_s \end{pmatrix}.$$

Let $f_t$ is the estimated $f$ using available information, i.e., $\{x_s, y_s, \boldsymbol{w}_s\}_{s=1}^t$. To get best linear unbiased estimator for $\boldsymbol{\beta}^\star$, we use the following equation from above matrix,

$$\left( \sum_{s=1}^t (\boldsymbol{w}_s - \boldsymbol{g}_s)^\top \varphi(x_s) \right) f_t + \left( \sum_{s=1}^t (\boldsymbol{w}_s - \boldsymbol{g}_s)^\top (\boldsymbol{w}_s - \boldsymbol{g}_s) \right) \hat{\boldsymbol{\beta}}_t = \sum_{s=1}^t (\boldsymbol{w}_s - \boldsymbol{g}_s) y_s$$

$$\implies \left( \sum_{s=1}^t (\boldsymbol{w}_s - \boldsymbol{g}_s)^\top (\boldsymbol{w}_s - \boldsymbol{g}_s) \right) \hat{\boldsymbol{\beta}}_t = \sum_{s=1}^t (\boldsymbol{w}_s - \boldsymbol{g}_s) y_s - \sum_{s=1}^t (\boldsymbol{w}_s - \boldsymbol{g}_s)^\top \left( \varphi(x_s)^\top f_t \right)$$

$$\implies \hat{\boldsymbol{\beta}}_t = \left( \sum_{s=1}^t (\boldsymbol{w}_s - \boldsymbol{g}_s)^\top (\boldsymbol{w}_s - \boldsymbol{g}_s) \right)^{-1} \sum_{s=1}^t (\boldsymbol{w}_s - \boldsymbol{g}_s) \left( y_s - \varphi(x_s)^\top f_t \right)$$

$$\implies \hat{\boldsymbol{\beta}}_t = \left( \sum_{s=1}^t (\boldsymbol{w}_s - \boldsymbol{g}_s)^\top (\boldsymbol{w}_s - \boldsymbol{g}_s) \right)^{-1} \sum_{s=1}^t (\boldsymbol{w}_s - \boldsymbol{g}_s) \left( y_s - f_t(x_s) \right)$$

Using definition $f_t(x_s) = \varphi(x_s)^\top f_t$, $\boldsymbol{W}_t = \begin{pmatrix} \boldsymbol{w}_1 - \boldsymbol{g}_s \\ \vdots \\ \boldsymbol{w}_t - \boldsymbol{g}_s \end{pmatrix}$, and $\boldsymbol{Y}_t = \begin{pmatrix} y_1 - f_t(x_1) \\ \vdots \\ y_t - f_t(x_t). \end{pmatrix}$, we get

$$\implies \hat{\boldsymbol{\beta}}_t = (\boldsymbol{W}_t^\top \boldsymbol{W}_t)^{-1} \boldsymbol{W}_t^\top \boldsymbol{Y}_t. \qquad \square$$

Since the reward and its auxiliary feedback observations are functions of the selected action, we can not directly use the control variate theory due to parameterized mean values of the reward and its auxiliary feedback. To overcome this challenge, we centered the observations by its function value and defined new centered variables as follows:

$$y_s^c = y_s - f(x_s), \quad \boldsymbol{w}_s^c = \boldsymbol{w}_s - \boldsymbol{g}_s, \quad \text{and} \quad z_{s,q}^c = z_{s,q} - f(x_s).$$

In our setting, these centered variables ($y_s^c$, $\boldsymbol{w}_s^c$, and $z_s^c$,) follow zero mean Gaussian distributions with variance $\sigma^2$, $\boldsymbol{\sigma}_w^2 = (\sigma_{w,1}^2, \ldots, \sigma_{w,q}^2)$, and $(1 - \rho^2)\sigma^2$, respectively.

**Theorem 1.** *Let $t > q + 2 \in \mathbb{N}$. If $\hat{\boldsymbol{\beta}}_t$ as defined in Lemma 1 is used to compute hybrid reward $z_{s,q}$ for any $s \le t \in \mathbb{N}$, then $\mathbb{E}[z_{s,q}] = f(x_s)$ and $\mathbb{V}(z_{s,q}) = \left( 1 + \frac{q}{t-q-2} \right) (1 - \rho^2)\sigma^2$.*

*Proof.* The sequence $(y_s^c, \boldsymbol{w}_s^c)_{s=1}^t$ is an IID sequence and follows a Gaussian distribution with mean 0. We now define $z_{s,q}^c = y_s^c - \boldsymbol{w}_s^c \boldsymbol{\beta} = y_s^c - (\boldsymbol{w}_s - \boldsymbol{g}_s)\boldsymbol{\beta}$, which can be re-written as $y_{s,q}^c = z_s^c + (\boldsymbol{w}_s - \boldsymbol{g}_s)\boldsymbol{\beta}$. Let $f_t$ be the estimated $f$ using available information, i.e., $\{x_s, y_s, \boldsymbol{w}_s\}_{s=1}^t$

and hence we can write estimated $z_{s,q}$ as $\hat{z}_{s,q} = f_t(x_s)$ and hence $\hat{z}^c_{s,q} = f_t(x_s) - f(x_s)$. Now, adapting Eq. (6) to our setting and replacing estimated mean in $Y_t$ by $\hat{z}^c_{s,q}$, $s^{\text{th}}$ value of $Y_t$ is $y^c_s - \hat{z}^c_{s,q} = y_s - f(x_s) - (f_t(x_s) - f(x_s)) = y_s - f_t(x_s)$. With these manipulations, we get the following best linear unbiased estimator for $\boldsymbol{\beta}^\star$:

$$\hat{\boldsymbol{\beta}}_t = (\boldsymbol{W}_t^\top \boldsymbol{W}_t)^{-1} \boldsymbol{W}_t^\top \boldsymbol{Y}_t,$$

which is the same as defined in Lemma 1.

By adapting Fact 4 for a single sample (i.e., $z_{s,q}$) while using $\hat{\boldsymbol{\beta}}_t$ to define hybrid reward, we have

$$\mathbb{E}\left[z^c_{s,q}\right] = 0 \text{ and}$$

$$\mathbb{V}\left(z^c_{s,q}\right) = \left(1 + \frac{q}{t-q-2}\right)(1-\rho^2)\mathbb{V}\left(y^c_s\right),$$

By extending the definition of $\mathbb{E}\left[z^c_{s,q}\right]$ we have,

$$\mathbb{E}\left[z_{s,q} - f(x_s)\right] = 0 \implies \mathbb{E}\left[z_{s,q}\right] - f(x_s) = 0 \implies \mathbb{E}\left[z_{s,q}\right] = f(x_s)$$

This proofs the hybrid reward is an unbiased estimator of reward.

Since variance is invariant to constant change, we have

$$\begin{aligned}
\mathbb{V}\left(z_{s,q}\right) &= \mathbb{V}\left(z_{s,q} - f(x_s)\right) \\
&= \mathbb{V}\left(z^c_{s,q}\right) \\
&= \left(1 + \frac{q}{t-q-2}\right)(1-\rho^2)\mathbb{V}\left(y^c_s\right) \\
&= \left(1 + \frac{q}{t-q-2}\right)(1-\rho^2)\mathbb{V}\left(y_s - f(x_s)\right) \\
&= \left(1 + \frac{q}{t-q-2}\right)(1-\rho^2)\mathbb{V}\left(y_s\right).
\end{aligned}$$

Since $\mathbb{V}\left(y_s\right) = \sigma^2$, we have $\mathbb{V}\left(z_{s,q}\right) = \left(1 + \frac{q}{t-q-2}\right)(1-\rho^2)\sigma^2$. $\qquad\square$

**Lemma 2.** *Let $t > q + 2 \in \mathbb{N}$, $e$ is the sampling strategy, and $f_t$ be the estimate of function $f$ at the end of round $t$ which uses $\{x_s, y_s, \boldsymbol{w}_s\}_{s=1}^t$. Then, the best linear unbiased estimator of $\boldsymbol{\beta}^\star_e$ is*

$$\hat{\boldsymbol{\beta}}_{e,t} = (\boldsymbol{W}_t^\top \boldsymbol{W}_t \circ \boldsymbol{F}_e)^{-1}\left(\text{diag}\left(\boldsymbol{F}_e\right) \circ \boldsymbol{W}_t^\top \boldsymbol{Y}_t\right),$$

*where $\boldsymbol{W}_t$ is a $t \times q$ matrix whose $s^{\text{th}}$ row is $\boldsymbol{w}_s - \boldsymbol{g}_{e,s}$ and $\boldsymbol{Y}_t = (y_1 - f_t(x_1), \ldots, y_t - f_t(x_t))$.*

*Proof.* Recall the $s^{\text{th}}$ hybrid reward defined in Eq. (3) using sampling strategy $e$ as $z^e_{e,s,q} = y_s - (\boldsymbol{w}_s - \boldsymbol{g}_{e,s})\boldsymbol{\beta}_e$, which can be re-written as $y_s = z^e_{e,s,q} + (\boldsymbol{w}_s - \boldsymbol{g}_{e,s})\boldsymbol{\beta}_e$. Following similar steps as of Lemma 1, we can re-write the above equation as $y_s = f^\top \varphi(x_s) + (\boldsymbol{w}_s - \boldsymbol{g}_{e,s})\boldsymbol{\beta}_e + \varepsilon_{z,s}$.

Using $\boldsymbol{W}_{e,t} = \begin{pmatrix} \boldsymbol{w}_1 - \boldsymbol{g}_{e,s} \\ \vdots \\ \boldsymbol{w}_t - \boldsymbol{g}_{e,s} \end{pmatrix}$, and $\boldsymbol{Y}_t = \begin{pmatrix} y_1 - f_t(x_1) \\ \vdots \\ y_t - f_t(x_t). \end{pmatrix}$, we get $\hat{\boldsymbol{\beta}}_{e,t} = (\boldsymbol{W}_{e,t}^\top \boldsymbol{W}_{e,t})^{-1}\boldsymbol{W}_{e,t}^\top \boldsymbol{Y}_t$.

From Appendix D and E of Gorodetsky et al. (2020), we have $\boldsymbol{W}_{e,t}^\top \boldsymbol{W}_{e,t} = \boldsymbol{W}_t^\top \boldsymbol{W}_t \circ \boldsymbol{F}_e$ and $\boldsymbol{W}_{e,t}^\top \boldsymbol{Y}_t = \text{diag}\left(\boldsymbol{F}_e\right) \circ \boldsymbol{W}_t^\top \boldsymbol{Y}_t$. Using these two equality, we have

$$\hat{\boldsymbol{\beta}}_{e,t} = (\boldsymbol{W}_t^\top \boldsymbol{W}_t \circ \boldsymbol{F}_e)^{-1}(\text{diag}\left(\boldsymbol{F}_e\right) \circ \boldsymbol{W}_t^\top \boldsymbol{Y}_t). \qquad\square$$

**Theorem 3.** *Let $t > q + 2 \in \mathbb{N}$ and $e$ is the sampling strategy. If $\hat{\boldsymbol{\beta}}_{e,t}$ as defined in Lemma 2 is used to compute hybrid reward $z_{e,s,q}$ for any $s \leq t \in \mathbb{N}$, then $\mathbb{E}\left[z_{e,s,q}\right] = f(x_s)$ and $\mathbb{V}\left(z_{e,s,q}\right) = \left(1 + \frac{a(e)q}{t-q-2}\right)(1-\rho_e^2)\sigma^2$, where $a(IS) = 1$, $a(MF) = \frac{r-1}{r}$ if $r_i = r$, $\forall i \in \{1, 2, \ldots, q\}$ when using MF sampling strategy for estimating auxiliary feedback functions.*

*Proof.* The proof follows the similar steps as Theorem 1 except we adapt the part (b.) of Theorem 4 from Pham and Gorodetsky (2022) instead of using Fact 4 to show the variance reduction when sampling strategy (IS or MF) is used for estimating auxiliary feedback functions. $\qquad\square$

## A.2 Unbiased estimate of variance

Consider the following regression problem with target variable $y_s$, which is defined as follows:

$$y_s = \mu + (\boldsymbol{w}_s - \boldsymbol{\omega})\boldsymbol{\beta} + \varepsilon_{z,s}.$$

where $\varepsilon_{z,1}, \ldots, \varepsilon_{z,t}$ are IID and have zero mean Gaussian noise with variance $(1 - \rho^2)\sigma^2$. Let

$$\overline{\boldsymbol{Y}}_t = \begin{pmatrix} y_1 \\ \vdots \\ y_t \end{pmatrix}, \ \overline{\boldsymbol{W}}_t = \begin{pmatrix} 1 & \boldsymbol{w}_1 - \boldsymbol{\omega} \\ \vdots & \vdots \\ 1 & \boldsymbol{w}_t - \boldsymbol{\omega} \end{pmatrix}, \ \boldsymbol{\gamma} = \begin{pmatrix} \mu \\ \boldsymbol{\beta} \end{pmatrix}, \ \text{and } \boldsymbol{\varepsilon}_{z,t} = \begin{pmatrix} \varepsilon_{z,1} \\ \vdots \\ \varepsilon_{z,t} \end{pmatrix}.$$

Now, using Fact 1, we have $\mathbb{V}(\hat{\mu}_{z,t}) = \sigma^2 (\overline{\boldsymbol{W}}_t^\top \overline{\boldsymbol{W}}_t)^{-1}_{11}$, where $(\boldsymbol{Y}^\top \boldsymbol{Y})^{-1}_{11}$ is the upper left most element of matrix $(\boldsymbol{Y}^\top \boldsymbol{Y})^{-1}$ (Schmeiser, 1982). Then after $t$ observations, the unbiased variance estimator of $\mathbb{V}(\hat{\mu}_{z,t})$ is given by

$$\hat{\nu}_{z,t} = \hat{\sigma}_{z,t}^2 (\overline{\boldsymbol{W}}_t^\top \overline{\boldsymbol{W}}_t)^{-1}_{11},$$

where $\hat{\sigma}_{z,t}^2 = \frac{1}{t-q-1} \sum_{s=1}^{t} (y_s - \hat{\mu}_{z,t})^2$ (Nelson, 1990), which is also an unbiased variance estimator of $\sigma^2$ (from Fact 2). Further, $\hat{\nu}_{z,t}$ is also an unbiased estimator of $\mathbb{V}(\hat{\mu}_{z,t})$, i.e., $\mathbb{E}[\hat{\nu}_{z,t}] = \mathbb{V}(\hat{\mu}_{z,t})$ (Nelson, 1990, Theorem 1). We can adapt this approach to our setting. However when noise variance $(\sigma)$ is unknown, computing $(\overline{\boldsymbol{W}}_t^\top \overline{\boldsymbol{W}}_t)^{-1}_{11}$ may not be possible to general function $f$ as $\varphi$ function may not be known. Though the setting in which $(\overline{\boldsymbol{W}}_t^\top \overline{\boldsymbol{W}}_t)^{-1}_{11}$ can be computed, we have to use the upper bound of variance to construct the confidence bound for reward function $f$ as random sample variance estimate can be small and leads to invalid confidence bounds. Given $t$ observations, the upper bound of the sample variance is given by $\bar{\nu}_{z,t} = \frac{(t-2)\hat{\nu}_{z,t}}{\chi^2_{1-\delta,t}}$, where $\chi^2_{1-\delta,t}$ denotes $100(1-\delta)^{\text{th}}$ percentile value of the chi-squared distribution with $t - 2$ degrees of freedom.

## A.3 Missing proofs related to regret analysis

**Theorem 2.** *With a probability of at least $1 - 2\delta$, the instantaneous regret of* OFUL-AF *in round $t$ is*

$$r_t(\text{OFUL-AF}) \leq 2 \left( \alpha_t^\sigma + \lambda^{1/2} S \right) \|x_t\|_{\overline{V}_t^{-1}},$$

*where* $\alpha_t^\sigma = \sqrt{\min(\sigma^2, \bar{\nu}_{z,t-1})} \, \alpha_t$, $\|\theta^\star\|_2 \leq S$, *and* $\alpha_t = \sqrt{d \log\left(\frac{1+tL^2/\lambda}{\delta}\right)}$. *For $t > q + 2$ and* $\bar{\nu}_{z,t} < \sigma^2$, $\mathbb{E}[r_t(\text{OFUL-AF})] \leq \widetilde{O}\left( \left(\frac{(t-3)(1-\rho^2)}{t-q-3}\right)^{\frac{1}{2}} r_t(\text{OFUL}) \right)$. *Here, $\widetilde{O}$ hides constant terms.*

*Proof.* When only observed rewards are used for estimating underlying unknown parameters in the linear bandit setting, i.e., $\hat{\theta}_t = \overline{V}_t^{-1} \sum_{s=1}^{t} x_s y_s$, then with probability $1 - \delta$, the confidence bound (Abbasi-Yadkori et al., 2011, Theorem 1) is

$$\left\|\hat{\theta}_t - \theta^\star\right\|_{\overline{V}_t} \leq \sigma \sqrt{d \log\left(\frac{1 + tL^2/\lambda}{\delta}\right)} + \lambda^{1/2} S, \tag{7}$$

where $\sigma^2$ is the variance of observed rewards given action (since the noise variance is $\sigma^2$). To ensure the performance of OFUL-AF is as good as OFUL, we only used hybrid reward samples for estimation when the upper bound on the variance of hybrid rewards is smaller than the variance of rewards, i.e., $\bar{\nu}_{z,t-1} < \sigma^2$. At the beginning of round $t$, the variance upper bound of hybrid rewards is computed using $t - 1$ observations and given by $\bar{\nu}_{z,t-1} = \frac{(t-2)\hat{\nu}_{z,t-1}}{\chi^2_{1-\delta,t}}$, where $\hat{\nu}_{z,t-1}$ is an unbiased sample variance estimate of hybrid rewards using $t - 1$ observations and $\chi^2_{1-\delta,t}$ (implying the variance upper bound holds with at least probability of $1 - \delta$) denotes $100(1-\delta)^{\text{th}}$ percentile value of the chi-squared distribution with $t - 2$ degrees of freedom. When $\bar{\nu}_{z,t} < \sigma^2$, we replace rewards $\{y_s\}_{s=1}^{t}$ with its respective hybrid rewards, i.e., $\{z_{s,q}\}_{s=1}^{t}$ to estimate underlying parameter and use it for next round. After using hybrid rewards to estimate the unknown parameter, we replace

$\sigma^2$ in Eq. (7) with the variance upper bound of hybrid rewards. Then we get the following upper bound which holds with a probability of $1 - 2\delta$.

$$\left\| \hat{\theta}_t - \theta^\star \right\|_{\bar{V}_t} \leq \sqrt{\min\left(\sigma^2, \bar{\nu}_{z,t-1}\right)} \sqrt{d \log\left(\frac{1 + tL^2/\lambda}{\delta}\right)} + \lambda^{1/2} S$$

$$= \sqrt{\min\left(\sigma^2, \bar{\nu}_{z,t-1}\right)} \alpha_t + \lambda^{1/2} S$$

$$\implies \left\| \hat{\theta}_t - \theta^\star \right\|_{\bar{V}_t} = \alpha_t^\sigma + \lambda^{1/2} S,$$

where $\alpha_t^\sigma = \sqrt{\min\left(\sigma^2, \bar{\nu}_{z,t-1}\right)} \alpha_t$ and $\alpha_t = \sqrt{d \log\left(\frac{1 + tL^2/\lambda}{\delta}\right)}$.

Let action $x_t$ be selected in the round $t$. Then, the instantaneous regret is given as follows:

$$r_t = \max_{x \in \mathcal{X}} x^\top \theta^\star - x_t^\top \theta^\star$$

$$= x^{\star\top} \theta^\star - x_t^\top \theta^\star \qquad \left(\text{as } x^\star = \max_{x \in \mathcal{X}} x^\top \theta^\star\right)$$

$$= (x^\star - x_t)^\top \theta^\star$$

$$= (x^\star - x_t)^\top \theta^\star + (x^\star - x_t)^\top \hat{\theta}_t - (x^\star - x_t)^\top \hat{\theta}_t$$

$$= (x^\star - x_t)^\top \hat{\theta}_t - (x^\star - x_t)^\top (\hat{\theta}_t - \theta^\star).$$

A sub-optimal action is only selected when its upper confidence bound is larger than the optimal action. Then, if $\left\| \hat{\theta}_t - \theta^\star \right\|_{\bar{V}_t} = \alpha_t^\sigma + \lambda^{1/2} S$, then we have

$$r_t \leq \alpha_t \|x_t\|_{\bar{V}_t^{-1}} - \alpha_t \|x^\star\|_{\bar{V}_t^{-1}} - (x^\star - x_t)^\top (\hat{\theta}_t - \theta^\star)$$

$$\leq \alpha_t \|x_t\|_{\bar{V}_t^{-1}} - \alpha_t \|x^\star\|_{\bar{V}_t^{-1}} - \|x^\star - x_t\|_{\bar{V}_t^{-1}} \left\| \hat{\theta}_t - \theta^\star \right\|_{\bar{V}_t}$$

$$\leq \alpha_t \|x_t\|_{\bar{V}_t^{-1}} - \alpha_t \|x^\star\|_{\bar{V}_t^{-1}} + \alpha_t \|x^\star - x_t\|_{\bar{V}_t^{-1}}$$

$$= \alpha_t(\|x_t\|_{\bar{V}_t^{-1}} - \|x^\star\|_{\bar{V}_t^{-1}} + \|x^\star - x_t\|_{\bar{V}_t^{-1}})$$

$$\leq \alpha_t(\|x_t\|_{\bar{V}_t^{-1}} - \|x^\star\|_{\bar{V}_t^{-1}} + \|X_{t,a_t^\star}\|_{\bar{V}_t^{-1}} + \|x_t\|_{\bar{V}_t^{-1}})$$

$$= 2\alpha_t \|x_t\|_{\bar{V}_t^{-1}}$$

$$\implies r_t \leq 2(\alpha_t^\sigma + \lambda^{1/2} S) \|x_t\|_{\bar{V}_t^{-1}}.$$

Let $\boldsymbol{X}_t = \{x_s\}_{s=1}^t$. For $t > q + 2$ and $\bar{\nu}_{z,t} < \sigma^2$, the expected instantaneous regret of OFUL-AF is

$$\mathbb{E}\left[r_t(\text{OFUL-AF})\right] \leq \mathbb{E}\left[2(\alpha_t^\sigma + \lambda^{1/2} S) \|x_t\|_{\bar{V}_t^{-1}}\right]$$

$$= \mathbb{E}\left[2(\sqrt{\bar{\nu}_{z,t}} \alpha_t + \lambda^{1/2} S) \|x_t\|_{\bar{V}_t^{-1}}\right]$$

$$= 2\mathbb{E}\left[\mathbb{E}\left[(\sqrt{\bar{\nu}_{z,t}} \alpha_t + \lambda^{1/2} S) \|x_t\|_{\bar{V}_t^{-1}} | \boldsymbol{X}_t\right]\right]$$

$$= 2\mathbb{E}\left[\alpha_t \|x_t\|_{\bar{V}_t^{-1}} \mathbb{E}\left[\sqrt{\bar{\nu}_{z,t}} | \boldsymbol{X}_t\right] + \lambda^{1/2} S \|x_t\|_{\bar{V}_t^{-1}}\right]$$

$$\leq 2\alpha_t \|x_t\|_{\bar{V}_t^{-1}} \mathbb{E}\left[\mathbb{E}\left[\sqrt{\frac{(t-2)\hat{\nu}_{z,t-1}}{\chi^2_{1-\delta,t}}} | \boldsymbol{X}_t\right]\right] + 2\lambda^{1/2} S \|x_t\|_{\bar{V}_t^{-1}}$$

$$= 2\alpha_t \|x_t\|_{\bar{V}_t^{-1}} \sqrt{\frac{(t-2)}{\chi^2_{1-\delta,t}}} \mathbb{E}\left[\sqrt{\hat{\nu}_{z,t-1}}\right] + 2\lambda^{1/2} S \|x_t\|_{\bar{V}_t^{-1}}.$$

Since $\hat{\nu}_{z,t-1}$ is an unbiased estimator of the sample variance of hybrid rewards, $\mathbb{E}\left[\hat{\nu}_{z,t-1}\right] = \mathbb{V}\left(z_{s,q}\right)$ for $s \in \{1, \ldots, t\}$. Using Theorem 1, we have $\mathbb{V}\left(z_{s,q}\right) = \left(1 + \frac{q}{t-q-3}\right)(1 - \rho^2)\sigma^2 =$

$\left(\frac{(t-3)(1-\rho^2)}{t-q-3}\right)\sigma^2$ as $t^{\text{th}}$ observation is not available at the beginning of the round $t$. With increasing $t$, $C_t = \sqrt{\frac{(t-2)}{\chi^2_{1-\delta,t}}}$ tends to 1. With all these observations, we have

$$\mathbb{E}\left[r_t(\text{OFUL-AF})\right] \leq 2C_t \left(\frac{(t-3)(1-\rho^2)}{t-q-3}\right)^{\frac{1}{2}} \sigma\alpha_t \|x_t\|_{\overline{V}_t^{-1}} + 2\lambda^{1/2}S \|x_t\|_{\overline{V}_t^{-1}}$$

$$= 2\left(C_t\left(\frac{(t-3)(1-\rho^2)}{t-q-3}\right)^{\frac{1}{2}} \sigma\alpha_t + \lambda^{1/2}S\right)\|x_t\|_{\overline{V}_t^{-1}}$$

Let $r_t(\text{OFUL})$ be the upper bound on instantaneous regret for OFUL algorithm, i.e., $r_t(\text{OFUL}) = 2\left(\sigma\alpha_t + \lambda^{1/2}S\right)\|x_t\|_{\overline{V}_t^{-1}}$. Then, we have

$$\mathbb{E}\left[r_t(\text{OFUL-AF})\right] \leq 2C_t \left(\frac{(t-3)(1-\rho^2)}{t-q-3}\right)^{\frac{1}{2}} \left(\sigma\alpha_t + \lambda^{1/2}S\right)\|x_t\|_{\overline{V}_t^{-1}}$$

$$+ 2\left(1 - C_t\left(\frac{(t-3)(1-\rho^2)}{t-q-3}\right)^{\frac{1}{2}}\right)\lambda^{1/2}S \|x_t\|_{\overline{V}_t^{-1}}$$

$$\leq C_t \left(\frac{(t-3)(1-\rho^2)}{t-q-3}\right)^{\frac{1}{2}} r_t(\text{OFUL})$$

$$+ 2\left(1 - C_t\left(\frac{(t-3)(1-\rho^2)}{t-q-3}\right)^{\frac{1}{2}}\right)\lambda^{1/2}S \|x_t\|_{\overline{V}_t^{-1}}.$$

$$\implies \mathbb{E}\left[r_t(\text{OFUL-AF})\right] \leq \widetilde{O}\left(\left(\frac{(t-3)(1-\rho^2)}{t-q-3}\right)^{\frac{1}{2}} r_t(\text{OFUL})\right). \qquad \square$$

**Theorem 4.** *Let $\mathfrak{A}$ be an AFC bandit algorithm with $|f_t^{\mathfrak{A}}(x) - f(x)| \leq \sigma h(x, \mathcal{O}_t) + l(x, \mathcal{O}_t)$ and $\bar{\nu}_{e,z,t}$ be the upper bound on sample variance of hybrid reward, whose value is set to $\sigma^2$ for $t \leq q+2$. Then, with a probability of at least $1 - 2\delta$, the instantaneous regret of $\mathfrak{A}$ after using hybrid rewards (named $\mathfrak{A}$-AF) for reward function estimation in round $t$ is*

$$r_t(\mathfrak{A}\text{-AF}) \leq 2\min(\sigma, (\bar{\nu}_{e,z,t})^{\frac{1}{2}})h(x, \mathcal{O}_t) + l(x, \mathcal{O}_t),$$

*where $e = \{IS, MF, KF\}$, and KF denotes the case where auxiliary functions are known. For $t > q+2$ and $\bar{\nu}_{e,z,t} < \sigma^2$, $\mathbb{E}\left[r_t(\mathfrak{A}\text{-AF})\right] \leq \widetilde{O}\left(\left(\left(\frac{t-(1-a(e))q-3}{t-q-3}\right)(1-\rho_e^2)\right)^{\frac{1}{2}} r_t(\mathfrak{A})\right)$, where $a(KF) = 1$.*

*Proof.* Let $\mathfrak{A}$ be an AFC bandit algorithm with $|f_t^{\mathfrak{A}}(x) - f(x)| \leq \sigma h(x, \mathcal{O}_t) + l(x, \mathcal{O}_t)$ and $\bar{\nu}_{e,z,t}$ be the upper bound on sample variance of hybrid reward. After $\mathfrak{A}$ uses hybrid rewards for estimating function $f$, then, with probability at least $1 - 2\delta$,

$$|f_t^{\mathfrak{A}}(x) - f(x)| \leq \min(\sigma, (\bar{\nu}_{e,z,t})^{\frac{1}{2}})h(x, \mathcal{O}_t) + l(x, \mathcal{O}_t) \qquad (8)$$

The proof follows similar steps as the first part of the proof of Theorem 2. The only key difference is the upper bound of variance of hybrid rewards, which depends on the underlying sampling strategy based on whether auxiliary functions are known or unknown. The upper bound on sample variance is given by $\bar{\nu}_{e,z,t} = \frac{(t-2)\hat{\nu}_{e,z,t-1}}{\chi^2_{1-\delta,t}}$, where $\hat{\nu}_{e,z,t-1}$ is an unbiased sample variance estimate of hybrid rewards using $t-1$ observations with sampling strategy $e$ and $\chi^2_{1-\delta,t}$ (implying the variance upper bound holds with at least probability of $1-\delta$) denotes $100(1-\delta)^{\text{th}}$ percentile value of the chi-squared distribution with $t-2$ degrees of freedom.

Let action $x_t$ be selected in the round $t$. Then, the instantaneous regret is given as follows:

$$r_t = \max_{x\in\mathcal{X}} f(x) - f(x_t) = f(x^\star) - f(x_t) \qquad\qquad (\text{as } x^\star = \max_{x\in\mathcal{X}} f(x))$$

$$\leq \left| f_t^{\mathfrak{A}}(x^\star) + \min(\sigma, (\bar{\nu}_{e,z,t})^{\frac{1}{2}}) h(x^\star, \mathcal{O}_t) + l(x^\star, \mathcal{O}_t) - f(x_t) \right|$$

$$\leq \left| f_t^{\mathfrak{A}}(x_t) + \min(\sigma, (\bar{\nu}_{e,z,t})^{\frac{1}{2}}) h(x_t, \mathcal{O}_t) + l(x_t, \mathcal{O}_t) - f(x_t) \right|$$

$$\leq \left| f_t^{\mathfrak{A}}(x_t) - f(x_t) \right| + \min(\sigma, (\bar{\nu}_{e,z,t})^{\frac{1}{2}}) h(x_t, \mathcal{O}_t) + l(x_t, \mathcal{O}_t)$$

$$\leq 2 \min(\sigma, (\bar{\nu}_{e,z,t})^{\frac{1}{2}}) h(x_t, \mathcal{O}_t) + l(x_t, \mathcal{O}_t),$$

in which the first and last inequalities have used the upper bound given in Eq. (8), and the second inequality follows because actions are selected using the upper confidence bounds. The remaining proof will follow the similar steps as the second part of Theorem 2 except using Theorem 3 instead of Theorem 1 for quantifying the variance reduction due to hybrid rewards when IS or MF sampling strategy is used for estimating auxiliary feedback function. $\qquad\square$

### A.4 Auxiliary feedback in contextual bandits

Many real-life applications have some additional information readily available for the learner before selecting an action, e.g., users' profile information is known to the online platform before making any recommendations. Such information is treated as contextual information in bandit literature, and the bandit problem having contextual information is refereed as contextual bandits (Li et al., 2010). Since the value of the reward function also depends on the context, the learner's goal is to use contextual information to select a better action.

We extend our results for the contextual bandits problem. In this setting, we assume that a learner has been given an action set denoted by $\mathcal{A}$. In round $t$, the environment generates a vector $\left( x_{t,a}, y_{t,a}, \{w_{t,a,i}\}_{i=1}^{q} \right)$ for each action $a \in \mathcal{A}$. Here, $x_{t,a}$ is the context-action $d$-dimensional feature vector of observed context in round $t$ and action $a$, $y_{t,a}$ is the stochastic reward received for context-action pair $x_{t,a}$, and $w_{t,a,i}$ is the $i^{\text{th}}$ auxiliary feedback associated with the reward $y_{t,a}$. We assume that the reward is a function of the context-action pair $x_{t,a}$, which is given as $y_{t,a} = f(x_{t,a}) + \varepsilon_t$, where $f : \mathbb{R}^d \to \mathbb{R}$ is an unknown function and $\varepsilon_t$ is a zero-mean Gaussian noise with variance $\sigma^2$. The auxiliary feedback is also assumed to be a function of the context-action pair $x_{t,a}$, given as $W_{t,a,i} = g_i(x_{t,a}) + \varepsilon_{t,i}^w$, where $g_i : \mathbb{R}^d \to \mathbb{R}$ and $\varepsilon_{t,i}^w$ is a zero-mean Gaussian noise with variance $\sigma_w^2$. The correlation coefficient between reward and associated auxiliary feedback is denoted by $\rho$.

We denote the optimal action for a context observed in the round $t$ as $a_t^\star = \arg\max_{a \in \mathcal{A}} f(x_{t,a})$. The interaction between a learner and its environment is given as follows. At the beginning of round $t$, the environment generates a context, and then the learner selects an action $a_t$ from action set $\mathcal{A}$ for that context using past information of context-actions feature vector, observed rewards and its associated auxiliary feedback until round $t - 1$. After selecting action $a_t$, the learner receives a reward $(y_{t,a_t})$ with its associated auxiliary feedback and incurs a penalty (or instantaneous regret) $r_t$, where $r_t = f(x_{t,a_t^\star}) - f(x_{t,a_t})$. We aim to learn a sequential policy that selects actions to minimize the total penalty and evaluate the performance of such policy through *regret*, which is the sum of the penalty incurred by the learner. Formally, for $T$ contexts, the regret of a policy $\pi$ that selects action $a_t$ for a context observed in round $t$ is given by

$$\mathfrak{R}_T(\pi) = \sum_{t=1}^{T} \left( f(x_{t,a_t^\star}) - f(x_{t,a_t}) \right). \tag{9}$$

A policy $\pi$ is a good policy when it has sub-linear regret. This implies that the policy will eventually learn to recommend the best action for every context. Similar to the parameterized bandit problem case, we can use the existing contextual bandit algorithms, which are AFC bandit algorithms. Depending on the problem, an appropriate AFC contextual bandit algorithm is selected that uses hybrid rewards to estimate reward function. The smaller variance of hybrid rewards leads to tighter upper confidence bound of the unknown reward function and hence smaller regret.

### A.5 More details about experiments

To demonstrate the performance gain from using auxiliary feedback, we have considered three different bandit settings: linear bandits, linear contextual bandits, and non-linear contextual bandits. The details of the problem instance used in our experiments are as follows.

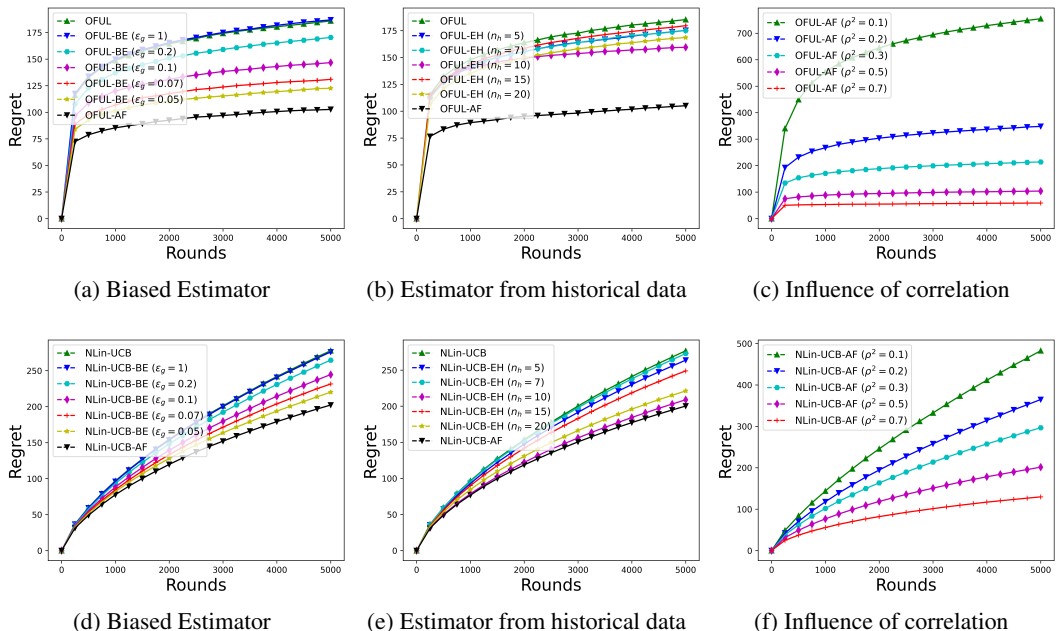

| (a) Biased Estimator | (b) Estimator from historical data | (c) Influence of correlation |
|---|---|---|
| (d) Biased Estimator | (e) Estimator from historical data | (f) Influence of correlation |

Figure 3: **Top row:** Experiment using linear bandits problem instance. **Bottom row:** Experiment using non-linear contextual bandits problem instance. **Left to right::** Regret vs. different biases (left figure), regret vs. number of historical samples of auxiliary feedback (middle figure), and regret vs. varying correlation coefficients of reward and its auxiliary feedback (right figure).

**Linear bandits:** We use a 5-dimensional space in which each sample is represented by $x = (x_1, \ldots, x_5)$, where the value of $x_j$ is restricted in $(-3, 3)$. We randomly select a 5-dimensional vector $\theta^\star$ with a unit norm whose each value is restricted in $(0, 1)$. In all linear bandits experiments, we use $\lambda = 0.01$, $L = 2.236$, $S = 1$, and $\delta = 0.05$. In round $t$, the reward for selected action $x_t$ is

$$y_t = v_t + w_t,$$

where $v_t = x_t^\top \theta_v^\star + \varepsilon_t^v$ and $w_t = x_t^\top \theta_w^\star + \varepsilon_t^w$. We set $\theta_v^\star = (0, \theta_2^\star, 0, \theta_4^\star, 0)$ and $\theta_w^\star = (\theta_1^\star, 0, \theta_3^\star, 0, \theta_5^\star)$. As we treat $w_t$ as auxiliary feedback, $\theta_w^\star$ may be assumed to be known in some experiments. The random noise $\varepsilon_t^v$ is zero-mean Gaussian noise with variance $\sigma_v^2$. Whereas $\varepsilon_t^w$ is also zero-mean Gaussian noise, but the variance is $\sigma_w^2$. We assumed that $\sigma^2 = \sigma_v^2 + \sigma_w^2$ is known, but not the $\sigma_v^2$ and $\sigma_w^2$. The default value of $\sigma_v^2 = 0.01$ and $\sigma_w^2 = 0.01$. It can be easily shown that the correlation coefficient of $y_t$ and $w_t$ is $\rho = \sqrt{\sigma_w^2/(\sigma_v^2 + \sigma_w^2)}$. We run each experiment for 5000 rounds.

**Linear contextual bandits:** We first generate a 2-dimensional synthetic dataset with 5000 data samples. Each sample is represented by $x = (x_1, x_2)$, where the value of $x_j$ is drawn uniformly at random from $(-1, 1)$. Our action set $\mathcal{A}$ has four actions: $\{(x_1, x_2), (x_1, -x_2), (-x_1, x_2), (-x_1, -x_2)\}$. We uniformly generate a $\theta^\star$ such that its norm is 1. In all experiments, the data samples are treated as contexts, and we use $\lambda = 0.01$, $L = 1.41$, $S = 1$, and $\delta = 0.05$. The observed reward for a context-action feature vector has two components. We treated one of the components as auxiliary feedback. In round $t$, the reward context-action feature vector $x_{t,a}$ is given as follows:

$$y_{t,a_t} = v_{t,a_t} + w_{t,a_t},$$

where $v_{t,a_t} = x_{t,a}^\top \theta_v^\star + \varepsilon_t^v$ and $w_{t,a_t} = x_{t,a}^\top \theta_w^\star + \varepsilon_t^w$. We set $\theta_v^\star = (0, \theta_2^\star, 0, \theta_4^\star)$ and $\theta_w^\star = (\theta_1^\star, 0, \theta_3^\star, 0)$. As we treat $w_{t,a_t}$ as auxiliary feedback, $\theta_w^\star$ is known for some experiments. The random noise $\varepsilon_t^v$ is zero-mean Gaussian noise with variance $\sigma_v^2$. Whereas $\varepsilon_t^w$ is also zero-mean Gaussian noise, but the variance is $\sigma_w^2$. We assumed that $\sigma^2 = \sigma_v^2 + \sigma_w^2$ is known, but not the $\sigma_v^2$ and $\sigma_w^2$. The default value of $\sigma_v^2 = 0.01$ and $\sigma_w^2 = 0.01$. It can be easily shown that the correlation coefficient of $y_{t,a}$ and $w_{t,a}$ is $\rho = \sqrt{\sigma_w^2/(\sigma_v^2 + \sigma_w^2)}$.

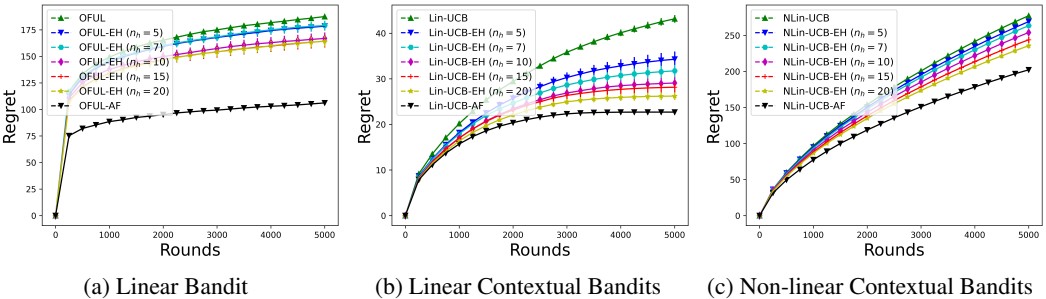

|  |  |  |
|:---:|:---:|:---:|
| (a) Linear Bandit | (b) Linear Contextual Bandits | (c) Non-linear Contextual Bandits |

Figure 4: Comparing regret vs. the number of historical samples of auxiliary feedback in different settings. In this experiment, historical samples for each run are randomly generated as compared to Fig. 2e, Fig. 3b, and Fig. 3e where history is kept fixed across the runs.

**Non-linear contextual bandits:** This problem instance is adapted from the linear contextual bandits problem instance. We first generate a 2-dimensional synthetic dataset with 5000 data samples. Each sample is represented by $x = (x_1, x_2)$, where the value of $x_j$ is drawn uniformly at random from $(-1, 1)$. We then use a polynomial kernel with degree 2 to have a non-linear transformation of samples. We removed (i.e., bias) the first (i.e., 1) and last value $(i.e., x_2^2)$ from the transformed samples, which reduced the dimensional of each transformed sample to 4 and represented as $(x_1, x_2, x_1^2, x_1 x_2)$, which is used as context. For this setting, the action set $\mathcal{A}$ has six actions: $\{(x_1, x_2, -x_1^2, -x_1 x_2), (x_1, -x_2, x_1^2, -x_1 x_2), (-x_1, x_2, x_1^2, -x_1 x_2), (x_1, -x_2, -x_1^2, x_1 x_2), (-x_1, x_2, -x_1^2, x_1 x_2), (-x_1, -x_2, x_1^2, x_1 x_2), \}$. We uniformly generate a $\theta^\star$ such that its norm is 1. In all experiments, we use $\lambda = 0.01$, $L = 2$, $S = 1$, and $\delta = 0.05$. The observed reward for a context-action feature vector has two components. We treated one of the components as auxiliary feedback. In round $t$, the reward context-action feature vector $x_{t,a}$ is given as follows:

$$y_{t,a_t} = v_{t,a_t} + w_{t,a_t},$$

where $v_{t,a_t} = x_{t,a}^\top \theta_v^\star + \varepsilon_t^v$ and $w_{t,a_t} = x_{t,a}^\top \theta_w^\star + \varepsilon_t^w$. We set $\theta_v^\star = (0, \theta_2^\star, 0, \theta_4^\star, 0, \theta_6^\star, 0, \theta_8^\star)$ and $\theta_w^\star = (\theta_1^\star, 0, \theta_3^\star, 0, \theta_5^\star, 0, \theta_7^\star, 0)$. As we treat $w_{t,a_t}$ as auxiliary feedback, $\theta_w^\star$ is known for some experiments. The random noise $\varepsilon_t^v$ is zero-mean Gaussian noise with variance $\sigma_v^2$. Whereas $\varepsilon_t^w$ is also zero-mean Gaussian noise, but the variance is $\sigma_w^2$. We assumed that $\sigma^2 = \sigma_v^2 + \sigma_w^2$ is known, but not the $\sigma_v^2$ and $\sigma_w^2$. The default value of $\sigma_v^2 = 0.01$ and $\sigma_w^2 = 0.01$. It can be easily shown that the correlation coefficient of $y_{t,a}$ and $w_{t,a}$ is $\rho = \sqrt{\sigma_w^2/(\sigma_v^2 + \sigma_w^2)}$.

**Regret with varying correlation coefficient:** As the correlation coefficient of reward and auxiliary feedback is $\rho = \sqrt{\sigma_w^2/(\sigma_v^2 + \sigma_w^2)}$, we varied $\sigma_v$ over the values $\{0.3, 0.2, 0.1528, 0.1, 0.0655\}$ to obtain problem instances with different correlation coefficient for all problem instances.

**Variance estimation:** Since the value of $\sigma^2$ is know in all our experiments, we directly estimate the correlation coefficient ($\rho$) as $\hat{\rho} = \text{Cov}(y, w)/(\sqrt{\mathbb{V}(w)}\sigma)$. Then, use it to set $\bar{\nu}_{e,z,t} = (1 - \hat{\rho}^2)\sigma^2$.

