# OpenReview forum: "Exploiting Correlated Auxiliary Feedback in Parameterized Bandits"
_NeurIPS.cc/2023/Conference — NeurIPS 2023 poster_

### Official Review · Reviewer_HUuF · 2023-07-03

**Soundness:** 3 good
**Presentation:** 3 good
**Contribution:** 3 good
**Rating:** 5
**Confidence:** 2

**Summary:**

In this paper, the authors studied a variant of the parameterized bandits problem where the learner has access to auxiliary feedback that is correlated with the observed reward. The authors proposed a method that leverages the auxiliary feedback to construct a reward estimator with more accurate confidence bounds, resulting in better regret bounds. The paper provides a characterization of the regret reduction in terms of the correlation coefficient between the reward and auxiliary feedback. Finally, they demonstrated the effectiveness of their method via numerical experiments.

**Strengths:**

1. The setting studied in this paper is interesting and realistically challenging, as it is often unclear how to use such auxiliary feedback for better online decision-making.
2. The paper is well organized. The theoretical results also appear sound.
3. The authors have provided a comprehensive literature review on existing research related to this work, and clearly explained the connection between their work and prior works in control variate theory.


**Weaknesses:**

It would be good if the authors could provide further discussions and clarifications on the following points:
1. The description of the algorithm (OFUL-AF) could be made clearer. Currently, it appears as a rephrasing of each step without clear connections to prior derivations. It would be helpful to explain how each step relates to the preceding derivations.
2. The procedure of selection of the number of auxiliary feedback $q$ in practice is not entirely clear to me. It'd be helpful to have more details regarding this point in addition to Remark 1.
3. The regret bound in Theorem 2 contains hidden constant terms. Could you elaborate more on the magnitude of these constant terms and their impact on the overall result?
4. The authors acknowledge that the actual form of the auxiliary feedback functions is typically unknown in practice, which I consider to be an important realistic challenge that needs to be dealt with. While Section 4 explores the effect of using estimated functions, I wonder how one can obtain an unbiased estimator for the auxiliary feedback functions in the first place. This is a challenging aspect that merits further explanation.
5. How should I comprehend the terms $a(e)$ and $\rho_e$ from Theorem 3? What do they each represent and do they constitute certain kind of tradeoff within the regret bound?
6. It'd be helpful to also see how the proposed model and method can be applied to certain real-world data. This is related to my point (4) above as I'm not entirely sure how one'd make sense of the auxiliary feedback function in practice.

**Questions:**

See weaknesses.

**Limitations:**

See weaknesses.

---

> ### Author Rebuttal · Authors · 2023-08-10
>
> Thank you for the detailed comments and suggestions. We have responded to each of the questions below.
>
> >### *1. The description of the algorithm (OFUL-AF) could be made clearer. Currently, it appears as a rephrasing of each step without clear connections to prior derivations. It would be helpful to explain how each step relates to the preceding derivations.*
>
> To improve the readability, we will add more explanations and connections to prior results while explaining each step of OFUL-AF in the revised version of the paper.
>
> >### *2. The procedure of selection of the number of auxiliary feedback $q$ in practice is not entirely clear to me. It'd be helpful to have more details regarding this point in addition to Remark 1.*
>
> The variance reduction given in Theorem 1 depends on different auxiliary feedback via two terms: $\frac{t-2}{t-q-2}$ and $\rho^2$ (defined in Line 142). As we increase the number of auxiliary feedback, $\rho^2$ will increase, leading to more variance reduction. However, simultaneously, the term $\frac{t-2}{t-q-2}$ will also increase, negating the gain in variance reduction. Therefore, it is recommended to use a small number of auxiliary feedback to avoid the degradation in variance reduction. One of the possible procedures for the selection of the number of auxiliary feedback is given in Lavenberg et al., 1982 (specifically on page 196).
>
> >### *3. The regret bound in Theorem 2 contains hidden constant terms. Could you elaborate more on the magnitude of these constant terms and their impact on the overall result?*
>
> There are two constants: one is the multiplicative constant $C_t$ (defined in Line 528 in the Appendix), whose value goes to $1$ as $t$ increases. Another is an additive constant (more details after Line 530 in the Appendix), which is comparatively negligible.
>
>
> >### *4. The authors acknowledge that the actual form of the auxiliary feedback functions is typically unknown in practice, which I consider to be an important realistic challenge that needs to be dealt with. While Section 4 explores the effect of using estimated functions, I wonder how one can obtain an unbiased estimator for the auxiliary feedback functions in the first place. This is a challenging aspect that merits further explanation.*
>
> Based on the application, a learner may have already collected auxiliary feedback (historical data) or may collect auxiliary feedback independent of the reward sample (e.g., having access to cheap low-fidelity simulations) or even collect additional samples of auxiliary feedback without a reward sample, e.g., an online food platform will record the food delivery time (auxiliary feedback) for every order but may not get a user rating (reward) for each order.
>
> In practice, one can use an appropriate variant of our proposed method depending on the need of their application (EH and BE variants shown in Fig. 2(a)-2(c)). However, it will be challenging to quantify the gain in variance reduction (as in Theorem 3) for these variants.
>
> >### *5. How should I comprehend the terms $a(e)$ and $\rho_e$ from Theorem 3? What do they each represent and do they constitute certain kind of tradeoff within the regret bound?*
>
> The term $a(e)$ comes due to different sampling methods used by IS and MF sampling strategies for estimating the unknown auxiliary functions. As given in Theorem 3, the value of $a(IS)$ (for IS sampling strategy) is 1, whereas it is $\frac{r-1}{r}$ for MF sampling strategy having $r_i = r$ for each auxiliary variable.
>
> The term $\rho_e$ denotes the multiple correlation coefficient of reward and its auxiliary feedback when using IS and MF sampling strategies. Its value depends on the sampling strategy (Line 231 for definition and Eqs. before Line 239).
>
> From the expression given in Line 282, smaller $a(e)$ and larger $\rho_e$ lead to smaller regret.
>
>
> >### *6. It'd be helpful to also see how the proposed model and method can be applied to certain real-world data. This is related to my point (4) above as I'm not entirely sure how one'd make sense of the auxiliary feedback function in practice.*
>
> Since it is common in bandit literature to measure the performance of bandits algorithms on synthetically generated data, we have also validated our method using different bandit instances. In the future, we will set up an elaborate experiment to demonstrate the effectiveness of our algorithms on real datasets.

---

> > ### Comment · Reviewer_HUuF · 2023-08-20
> >
> > Thank you for your response. My main remaining concern is about the assumed knowledge of the auxiliary feedback functions, or its unbiased estimator. Could you elaborate on how you would actually construct an unbiased estimator by using historical data or acquiring more samples of auxiliary feedback? Given the noisy environment one would usually face in practice, it'd be natural to expect the existence of bias. However, it seems that if the bias gets large the results can worsen quite a bit. Is there a way to theoretically quantify how much your regret might worsen given the extent of bias?
> >
> > Given my concern above, I also think experiments on real-world data would be a necessity here. I understand that it is common in bandits literature that synthetic experiments are adopted; nevertheless, given the authors' claim that auxiliary feedbacks are closely connected with real-life applications, while assumptions in this paper do not always apply to real-world settings (as discussed by reviewers above), it's important to understand what kind of modifications one might need to make to the proposed method for real-world scenarios.
> >
> > Due to the above reasons, I will keep my score as borderline.

---

> > > ### Author Response · Authors · 2023-08-21
> > > **Auxiliary feedback and experiments**
> > >
> > > Thank you for acknowledging our rebuttal. Here are our responses to your questions.
> > >
> > > > ### *My main remaining concern is about the assumed knowledge of the auxiliary feedback functions, or its unbiased estimator. Could you elaborate on how you would actually construct an unbiased estimator by using historical data or acquiring more samples of auxiliary feedback?*
> > >
> > > There are many applications (as mentioned in Lines 114-115) where auxiliary feedback can be constructed such that the corresponding auxiliary functions are known. However, when these auxiliary functions are unknown, we must estimate them to exploit the correlation between reward and its auxiliary feedback to get a better reward function estimator (i.e., estimator has tight confidence bounds).
> > >
> > > For constructing an unbiased estimator of auxiliary function, let us assume a linear relationship exists between features of action (or context-action) and corresponding auxiliary feedback, i.e., the auxiliary function is linear. When we have extra samples of auxiliary feedback (historical data or acquiring separately without the rewards samples), the solution of ordinary least square (for samples > numbers of features) gives an unbiased estimator, i.e., $\mathbb{E}[\hat{g}_m] = g$, where $\hat{g}_m$ is an ordinary least square estimator of the auxiliary function $g$ and uses $m$ samples. However, getting an unbiased auxiliary function estimator for any arbitrary non-linear auxiliary function may be difficult.
> > >
> > >
> > > > ### *Given the noisy environment one would usually face in practice, it'd be natural to expect the existence of bias. However, it seems that if the bias gets large the results can worsen quite a bit. Is there a way to theoretically quantify how much your regret might worsen given the extent of bias?*
> > >
> > > We agree with the reviewer's observation that the performance of our method will decline (i.e., regret increases, as shown in Fig. 2(d)) as bias in estimated auxiliary feedback increases. We have not theoretically quantified the relationship between regret and bias in estimated auxiliary feedback in this paper. To do this, one has to construct new confidence intervals for the reward function estimator with biased hybrid rewards (as biased auxiliary feedback will affect the hybrid rewards as defined in Eq. (3)).
> > >
> > >
> > > > ### *Given my concern above, I also think experiments on real-world data would be a necessity here. I understand that it is common in bandits literature that synthetic experiments are adopted; nevertheless, given the authors' claim that auxiliary feedbacks are closely connected with real-life applications, while assumptions in this paper do not always apply to real-world settings (as discussed by reviewers above), it's important to understand what kind of modifications one might need to make to the proposed method for real-world scenarios.*
> > >
> > > Our work is mainly theoretical, and it is the first to demonstrate how one can use correlated auxiliary feedback (whenever available) to improve the performance (i.e., minimize the regret) of parameterized bandit algorithms. To establish the performance gains analytically, we have assumed (apart from common assumptions used in parameterized bandit papers) that the estimators of the auxiliary functions are unbiased. We agree that our assumptions may not hold in every real-world setting, but this is also true for many bandit algorithms.
> > >
> > > Our experimental goal is to verify our theoretical results (using correlated auxiliary feedback leads to smaller regret than existing parameterized bandit algorithms) and different properties of our proposed method (e.g., variation in regret and correlation between reward and its auxiliary feedback and regret variation with numbers of auxiliary feedback [in rebuttal]). To illustrate this, we used synthetic problem instances as it is easier to verify our theoretical results. We agree with the reviewer that our method may not be directly used in practice, and one has to make appropriate changes to adapt our method to their problems.
> > >
> > > *We hope that our answers will improve your opinion of our work. If you have additional questions, we would be happy to answer them.*

---

### Official Review · Reviewer_jBfm · 2023-07-03

**Soundness:** 3 good
**Presentation:** 3 good
**Contribution:** 2 fair
**Rating:** 4
**Confidence:** 4

**Summary:**

This paper leverages the method of control variates to obtain reward estimates with smaller variance for contextual bandit algorithms, since smaller variance in reward estimation means tighter confidence bound estimation and therefore smaller regret. Estimation methods for both unknown and known auxiliar feedback functions are provided, where both theoretical and empirical results were provided to demonstrate the effectiveness of the proposed solution.

**Strengths:**

Leveraging available auxiliar feedback to improve reward estimation is an important and also meaningful approach for improving bandit algorithms. This paper provides a viable solution to realize the goal.

The provided solution is an extension of Verma and Hanawal’s NeurIPS’2021 work in multi-armed bandits, and the authors further extended it to contextual bandit problems, especially with the non-linear reward functions.


**Weaknesses:**

The means of using control variates to reduce the reward estimation variance is not a completely new idea, as the authors pointed out in the related work discussions. And the estimation techniques employed in this paper were also borrowed from prior works. For example, the results under known auxiliary feedback function is a straightforward extension of Verma and Hanawal’s NeurIPS’2021 work on top of OFUL’s analysis, and the results for the unknown auxiliary feedback function is also not super challenging (e.g., using existing estimation technique and assuming the reward variance is known). This to certain degree limits the novelty of this work.

The description for the unbiased linear estimator $\beta_e$ is unfortunately unclear and could be actually problematic. Based on the problem setup, my understanding is that every arm pull reveals all $q$ dimensions of the auxiliary feedback, in addition to the reward feedback. And the different sampling/partitioning methods, e.g., IS or MF, decide how to allocate those samples for estimating the reward function and auxiliary feedback function. Hence, the total number of samples for estimation at time $t$ is $t$, but each sample has $q+1$ dimensions. But in line 302, it states to maintain $r=2$, it needs to “getting one extra sample of auxiliary feedback in each round”. This seems to suggest the algorithm can pull another arm and only require the auxiliary feedback for free. If this is the case for algorithm design, it is an unfair advantage to the algorithm, comparing to the baseline bandit algorithms, as it can collect more information about the reward. Or in other words, why not further increase the ratio $r$ to better exploit this advantage? The EH variant of this algorithm further confirmed my understanding: it is assumed more observations about the auxiliary feedback are available.

The experiment settings were also overly simplified, which do not strongly support the advantages or demonstrate the limitations of the proposed solution. For example, there is only one dimension of auxiliary feedback. Given the algorithm’s theoretical performance depends on the dimension of auxiliary feedback, it is important to vary its dimension to investigate its practical impact.

Another factor should be mentioned is that the paper only addresses the finite arm setting (especially in the experiments), though in contextual bandit problems infinite arms with potentially adversarial context arrival is believed to be a more general setting. Otherwise, simple algorithms can already achieve satisfactory regret: for example, a greedy algorithm can obtain sublinear regret with sufficient context diversity and logarithmic regret is also achievable under stochastic context distributions. It would be meaningful to discuss how the developed algorithm can be extended to this more general and also more challenging environment.


**Questions:**

Since we are using linear regression for $\beta$ estimation, why is $W^\top_t W_t$ guaranteed to be invertible? Even when we have more samples than the number of auxiliary dimensions, i.e., $t>q+2$, the observations of auxiliary feedback might not span the entire $q$ dimensional space, depending on the distribution of selected arms.

Is it true that the algorithm is supposed to have free access to auxiliary feedback from any arm? If so, what presents the algorithm from extensively pulling all the arms to estimate their auxiliary feedback function, so as to get the most accurate estimation of those functions first?

The algorithm assumes the reward noise is known and use this quantity to control when to use which estimator. But in practice, we do not know the actual value of reward variance, and how could we use the proposed algorithm?


**Limitations:**

I do not find any concerns regarding the negative societal impact of this work.

---

> ### Author Rebuttal · Authors · 2023-08-10
>
> Thank you for the detailed comments and constructive feedback. We have responded to each of the questions below.
>
> > ### *The means of using control variates to reduce the reward estimation variance is not a completely new idea, as the authors pointed out in the related work discussions. And the estimation techniques employed in this paper were also borrowed from prior works. $\ldots$ This to certain degree limits the novelty of this work.*
>
> Our work is motivated by Verma and Hanawal (2021). Please check the **Main contributions** part of the global rebuttal for our novel contributions.
>
> > ### *$\ldots$ in line 302, it states to maintain $r=2$, it needs to “getting one extra sample of auxiliary feedback in each round”. This seems to suggest the algorithm can pull another arm and only require the auxiliary feedback for free. If this is the case for algorithm design, it is an unfair advantage to the algorithm, comparing to the baseline bandit algorithms, as it can collect more information about the reward. Or in other words, why not further increase the ratio $r$ to better exploit this advantage? The EH variant of this algorithm further confirmed my understanding: it is assumed more observations about the auxiliary feedback are available.*
>
> Depending on the applications, a learner may have already collected auxiliary feedback (historical data) or may collect auxiliary feedback independent of the reward sample (e.g., cheap low-fidelity simulations) or even collect additional samples of auxiliary feedback without a reward sample, e.g., an online food platform will record the food delivery time (auxiliary feedback) for every order but may not get a user rating (reward) for each order. In our experiments, we maintain $r=2$ to validate our theoretical result, but getting an extra auxiliary sample does not imply that we will play that arm again but ignore the reward sample. Our proposed method uses all observations with reward samples to estimate the reward function $f$. In practice, one can use an appropriate variant of our proposed method depending on the need of their application (EH and BE variants shown in Fig. 2(a)-2(c)). However, it will be challenging to quantify the exact gain in variance reduction for these variants.
>
>
> > ### *The experiment settings were also overly simplified, $\ldots$. Given the algorithm’s theoretical performance depends on the dimension of auxiliary feedback, it is important to vary its dimension to investigate its practical impact.*
>
> Our work is a theoretical work that quantifies the performance gain achieved by a parameterized bandit algorithm using auxiliary feedback correlated with reward samples. It is common in bandit literature to measure the performance of bandits algorithms on synthetically generated data. Therefore, we have also validated our method using different bandit instances. We have also added an experiment result with larger $q (={1, 2, 3, 4, 5}$) when the auxiliary functions are known. For more details, please check the attached pdf in the global rebuttal.
>
>
> > ### *Another factor should be mentioned is that the paper only addresses the finite arm setting (especially in the experiments), though in contextual bandit problems infinite arms with potentially adversarial context arrival is believed to be a more general setting. $\ldots$.*
>
> Our experiment results related to OFUL as a baseline linear bandit algorithm deals with infinite arms setting (different results are shown in Fig. 2(a), Fig3(a), Fig3(b), Fig3(c), Fig4(a)). The main challenge for extending our method to more general settings and challenging environments is incorporating correlated auxiliary feedback with reward samples to improve reward function estimation. Our proposed method and techniques will be a baseline for future work in more challenging settings.
>
>
> > ### *Since we are using linear regression for $\beta$ estimation, why is $W_tW_t^\top$ guaranteed to be invertible?$\ldots$.*
>
> The randomness in each auxiliary feedback is due to IID Gaussian noise, which is independent of actions and other auxiliary feedback. It makes auxiliary feedback vectors independent of each other, and hence $W_tW_t^\top$ is invertible if there are more than $q$ auxiliary feedback vectors. When auxiliary feedback vectors are not independent, we can add a condition (e.g., Line 7 of OFUL-AF) for calculating $\boldsymbol{\hat\beta}_t$ only if $W_tW_t^\top$ is an invertible matrix.
>
>
> > ### *Is it true that the algorithm is supposed to have free access to auxiliary feedback from any arm? If so, what presents the algorithm from extensively pulling all the arms to estimate their auxiliary feedback function, so as to get the most accurate estimation of those functions first?*
>
> We have kept $r$ constant in Theorem 3 to quantify the variance reduction. However, this may not be the case in practice, e.g., having enough historical data of auxiliary feedback can be used to get a reasonable estimate of an auxiliary function. Consider another example of a cheap low-fidelity simulator where an algorithm can get as many auxiliary feedback samples as needed to have a good estimate of the auxiliary function at the start. However, using this may lead to a variant of our method for which quantifying the variance reduction may be challenging.
>
> > ### *$\ldots$ in practice, we do not know the actual value of reward variance, and how could we use the proposed algorithm?*
>
> We agree with the reviewer that we do not know the actual value of reward variance in practice. However, the assumption of a known (upper bound of) reward variance is common in many bandit algorithms, e.g., OFUL, Lin-UCB, UCB-GLM, and IGP-UCB. Further, our method can be extended to bandit algorithms like VOFUL and VOFUL2 for problems with unknown variance.

---

> > ### Comment · Reviewer_jBfm · 2023-08-17
> >
> > I appreciate the authors' explanations in the rebuttal, which help me better understand the technical details. Still a few points to follow up.
> >
> > > “In our experiments, we maintain $r=2$ to validate our theoretical result, but getting an extra auxiliary sample does not imply that we will play that arm again but ignore the reward sample. Our proposed method uses all observations with reward samples to estimate the reward function $f$.”
> >
> > I am a bit confused by this explanation: Given $r$ is the ratio between the samples used in estimation $g$ vs, $f$, my understanding of $r=2$ is that at each time we will have one observation of reward, and two observations of auxiliary feedback. But how could we get that extra sample for auxiliary feedback without pulling one more arm? Correct me if I misunderstood.
> >
> > > “We have also added an experiment result with larger $q (q=1,..,5)$ when the auxiliary functions are known.”
> >
> > Very glad to find this new result to verify the effectiveness of having more auxiliary feedback, though the results were obtained under the simplest setting of known auxiliary functions.
> >
> > In addition, the authors mentioned several times of leveraging historical data with auxiliary feedback and also tested its EH variant in the experiments; but in this case, we should compare with bandit algorithms that leverage offline data (the reward part), such as the following
> >
> >  - Zhang, Chicheng, et al. "Warm-starting contextual bandits: Robustly combining supervised and bandit feedback." arXiv preprint arXiv:1901.00301 (2019).
> >
> > > “the assumption of a known (upper bound of) reward variance is common in many bandit algorithms”
> >
> > I totally agree that this is a common assumption in most bandit algorithm, but my original intent was to ask what’s its practical impact: in UCB-type algorithms, the assumed reward noise scales an algorithm’s regret; while in the proposed algorithm, it not only scales regret but also affects when the benefit of auxiliary feedback appears. Not sure if this complicates hyper-parameter tuning.

---

> > > ### Author Response · Authors · 2023-08-18
> > > **Auxiliary feedback and additional experiments with multiple unknown auxiliary functions**
> > >
> > > Thank you for acknowledging our rebuttal. Here are our responses to your questions.
> > >
> > > > ### *... my understanding of $r=2$ is that at each time we will have one observation of reward, and two observations of auxiliary feedback. But how could we get that extra sample for auxiliary feedback without pulling one more arm? ...*
> > >
> > > We agree with the reviewer that we can get the extra sample for auxiliary feedback only after pulling an arm. To clarify, here is how we do it in our experiments. As the auxiliary function $g$ is known in our experiments (Lines 624-627 in Appendix, where $g$ is parameterized by $\theta_w^\star$), we can generate samples of auxiliary feedback (without reward) for randomly selected actions. Therefore, we have two types of observations -- one has a reward and associated auxiliary feedback for the selected action, while another only has auxiliary feedback for the random action.
> > >
> > > It leads to the question, *Is getting additional auxiliary samples without a reward sample even possible?* The answer is *Yes*. There are many real-life applications where auxiliary feedback is observed but not the reward for selected actions. For example, the fool delivery platform may not get a user rating for each order but can record the delivery time for every order. We have discussed such scenarios in the global rebuttal under **Availability of Auxiliary feedback**. Note that one can get extra samples of auxiliary feedback, but each sample may have an associated cost. For example, one can get multiple samples from a low-fidelity simulation model. However, each sample will have a computational cost (which may be very small compared to a high-fidelity simulation model).
> > >
> > > > ### *... new result to verify the effectiveness of having more auxiliary feedback, though the results were obtained under the simplest setting of known auxiliary functions.*
> > >
> > > We have also run additional experiments with multiple unknown auxiliary functions. We use a problem instance with $5$ auxiliary functions having different noise standard deviations (i.e., $\sigma = \{0.1, 0.08, 0.07, 0.06, 0.02\}$) and one unknown function with noise standard deviation of $0.1$. We chose the auxiliary function with the largest noise standard deviation for $q=1$ case, the auxiliary function with the two largest noise standard deviations for $q=2$ case, and so on. We set the number of rounds $(T)$ to $1000$ and $r=2$ for IS and MF sampling-based algorithms. We repeated all our experiments $100$ times ($50$ times for Lin-UCB-IS and Lin-UCB-MF) and showed the average cumulative regret as defined in Eq. (1) with a 95% confidence interval in the following table.
> > >
> > > |Algorithm\No. of AF|$q=1$|$q=2$|$q=3$|$q=4$|$q=5$|
> > > |-|-|-|-|-|-|
> > > |Lin-UCB-EH ($n_h=10$)|39.545$\pm$0.802|30.282$\pm$1.051|4.346$\pm$0.267|4.306$\pm$0.164|4.465$\pm$0.182|
> > > |Lin-UCB-BE ($\epsilon_g=0.1$)|32.259$\pm$0.871|21.612$\pm$1.048|9.414$\pm$0.83|11.984$\pm$1.401|18.586$\pm$2.022|
> > > |Lin-UCB-IS|44.816$\pm$0.729|43.693$\pm$0.805|94.558$\pm$1.208|159.855$\pm$2.282|185.856$\pm$2.485|
> > > |Lin-UCB-MF|44.816$\pm$0.729|43.615$\pm$0.747|93.961$\pm$1.305|161.287$\pm$1.816|190.807$\pm$1.756|
> > >
> > > As expected, regret decreases as q increases initially, but then it increases and even worsens than the baseline for Lin-UCB-IS and Lin-UCB-MF for $q>2$. For reference, the average cumulative regret incurred by Lin-UCB for the same problem instance was **50.75 $\pm$ 0.435.**
> > > > ### *In addition, the authors mentioned several times of leveraging historical data with auxiliary feedback and also tested its EH variant in the experiments; but in this case, we should compare with bandit algorithms that leverage offline data (the reward part)...*
> > >
> > > When auxiliary functions are unknown, we need a good estimate of these functions to get maximum benefit from auxiliary feedback. One possible way to get a good estimated auxiliary function is to use historical data of auxiliary feedback for its estimation. We have not considered the problems where the historical data of reward is also available. It is an interesting direction to pursue in future, and one can start with techniques introduced in the suggested paper (Zhang et al., 2019).
> > >
> > > > ### *... in the proposed algorithm, it not only scales regret but also affects when the benefit of auxiliary feedback appears. Not sure if this complicates hyper-parameter tuning.*
> > >
> > > As we use a high probability upper bound on the noise variance of hybrid rewards, this upper bound may exceed the noise variance $(\sigma^2)$ of rewards. To ensure the proposed algorithm performs better than the baseline bandit algorithm, we only use hybrid reward when its estimated upper bound of noise variance is smaller than $\sigma^2$. Therefore, hyper-parameter tuning needs to be adjusted when the algorithm switches from using rewards to hybrid rewards for estimating the reward function.
> > >
> > > *We hope our answers will further improve your opinion of our work. If you have additional questions, we would be happy to address them.*

---

> > > > ### Comment · Reviewer_jBfm · 2023-08-19
> > > >
> > > > Many thanks for adding more experiment results, which look quite positive!
> > > >
> > > > Still two points I would like to clarify and emphasize:
> > > >
> > > > 1. I do not think the proposed algorithm here can leverage historical data only with auxiliary feedback, as it has to have reward to build the correlation between reward and auxiliary feedback. Am I right?
> > > >
> > > > 1. getting additional auxiliary feedback for free is an unfair advantage to classical bandit baselines that only leverage reward signal. But I totally agree that if one can leverage such advantage, it is a good merit!

---

> > > > > ### Author Response · Authors · 2023-08-21
> > > > > **Clarifications about auxiliary feedback**
> > > > >
> > > > > Here are our clarifications regarding the points you raised:
> > > > >
> > > > > > ### *I do not think the proposed algorithm here can leverage historical data only with auxiliary feedback, as it has to have reward to build the correlation between reward and auxiliary feedback. Am I right?*
> > > > >
> > > > > The algorithm variant that uses historical auxiliary feedback data (ended with 'EH') is not the paper's main contribution. Instead, this variant is a heuristic method and can only be used to get the estimated auxiliary functions when some historical auxiliary feedback data is available. We used this variant as a baseline method to compare IS and MF sampling-based algorithms for the bandit problems with unknown auxiliary functions.
> > > > >
> > > > > Please note that we do not need reward samples for estimating auxiliary functions as they are separately estimated and only use auxiliary feedback. However, we need both the reward and its auxiliary feedback to define hybrid rewards (as defined in Eq. (2) and Eq. (3)), which are then used to estimate the reward function.
> > > > >
> > > > >
> > > > > > ### *getting additional auxiliary feedback for free is an unfair advantage to classical bandit baselines that only leverage reward signal. But I totally agree that if one can leverage such advantage, it is a good merit!*
> > > > >
> > > > > We agree with the reviewer that getting additional auxiliary feedback for free is an unfair advantage over classical bandit algorithms that only use reward signals. However, there are many real-life applications (e.g., online platforms like online food delivery or e-commerce platform) where auxiliary feedback are freely available. To the best of our knowledge, no parameterized bandit algorithm uses auxiliary feedback.
> > > > > To fill this gap in bandit literature, this paper proposes a method to exploit correlated auxiliary feedback and theoretically quantifies the performance gain (in variance reduction and then improvement in regret upper bound) when auxiliary feedback are used.
> > > > >
> > > > >
> > > > >
> > > > > *We hope that we are able to address your concerns. If you have more questions, we would be happy to answer them.*

---

### Official Review · Reviewer_CBCn · 2023-07-04

**Soundness:** 3 good
**Presentation:** 4 excellent
**Contribution:** 3 good
**Rating:** 6
**Confidence:** 3

**Summary:**

This paper studies the parametrized bandit problem in which the learner observes auxiliary feedback together with the reward, and also correlated with the reward. It is motivated from the control variate approach in causal inference, the main difference is that in this paper, it extends the control variable theory to a setting where the "control variable" is parametrized by a function. It proposes a new bandit algorithm which replaces the original observed reward, with a version built upon the known/estimated control variate and studies the expected instantaneous regret compared with the original bandit algorithm.

**Strengths:**

- This paper studies a new bandit framework which differs from most of the prior work (such as side information, side observation, etc). In this new framework, the learner also observes the auxiliary feedback beyond the original reward. This setting is pretty relevant to lots of real-world scenarios, such as in food delivery platform, user rating might be revealed together with the delivery time, in recommendation platform, optimizing user like rate might be revealed together with the watch time, etc.

- The paper is very well-written and easy to follow. It is built upon the classical control variate theory, and extends it to the known function (of the control variate), then further down to estimated function setting. The method is solid, and the expected instantaneous regret is provided to validate the soundness of the proposed approach.

- I appreciate the additional efforts in the synthetic datasets to verify the empirical performance of the proposed method under various environment, such as the linear, linear contextual, non-linear contextual bandits, and study how the estimation of the control variate function as well as the correlation of the auxiliary feedback with the reward affect the final performance. These results facilitate the understanding.

**Weaknesses:**

- I am a little bit confused about the relationship of the variance reduction in terms of the number of auxiliary feedback being used. Under estimated $\beta$ and from Theorem 1, it seems the optimal number of auxiliary feedback being used is 1, which seems a little bit counter-intuitive, could the authors comment more on this aspect?

- In Section 4, under estimated auxiliary functions, the sampling strategy for estimating the auxiliary function seems very computationally inefficient, and this leaves much smaller sample size for estimating the original f function compared with classical bandit algorithms, especially when the number of auxiliary functions is large. Remark 2 also does not make that much of sense, when $r_i$ goes to infinity where we use most of the sample in auxiliary function estimate.

- The IS and MF sampling strategy are only listed in the method discussions, and for the experiments when $q=2$, it is hard to compare the pros and cons of them as they are equivalently being the same. Could the authors add more results for larger $q$ to showcase the effectiveness of the two sampling methods. A larger $q$ might also be helpful in understanding the effectiveness of having more control variates.

- I am not sure if there is any public dataset available to showcase the effectiveness of the method in more real-world scenarios. The current ablation experiments facilitate the understanding of the method, but adding real-world experiments would be much more convincing.


**Questions:**

- For Figure 2 (f), is it possible to add the performance for original Lin-UCB performance?
- It would be good to have an ablation study w.r.t the number of auxiliary functions.
- others listed in the Weakness section.

---

> ### Author Rebuttal · Authors · 2023-08-10
>
>
>
> Thank you for the detailed comments and suggestions. We have responded to each of the questions below.
>
> > ### *I am a little bit confused about the relationship of the variance reduction in terms of the number of auxiliary feedback being used. Under estimated $\beta$ and from Theorem 1, it seems the optimal number of auxiliary feedback being used is 1, which seems a little bit counter-intuitive, could the authors comment more on this aspect?*
>
> The variance reduction given in Theorem 1 depends on the auxiliary feedback via two terms: $\frac{t-2}{t-q-2}$ and $\rho^2$ (defined in Line 142). Setting $q=1$ will give the minimum value for $\frac{t-2}{t-q-2}$, but $\rho^2$ for $q=1$ will also be small as it only considers one auxiliary feedback, and hence maximum variance reduction will not be achieved. As we increase the number of auxiliary feedback, $\rho^2$ will increase, leading to more variance reduction. However, at the same time, the term $\frac{t-2}{t-q-2}$ will also increase, which can negate the variance reduction. Therefore, it is recommended to use a small number of auxiliary feedback (more in Remark 1).
>
>
>
> > ### *In Section 4, under estimated auxiliary functions, the sampling strategy for estimating the auxiliary function seems very computationally inefficient, and this leaves much smaller sample size for estimating the original f function compared with classical bandit algorithms, especially when the number of auxiliary functions is large. Remark 2 also does not make that much of sense, when $r_i$ goes to infinity where we use most of the sample in auxiliary function estimate.*
>
>
> In many real-world applications, a learner may have already collected auxiliary feedback (historical data) or collect auxiliary feedback independent of the reward sample (e.g., cheap low-fidelity simulations) or even collect additional samples of auxiliary feedback with no reward sample, e.g., an online food platform will not get a user rating (reward) for each order but will have a food delivery time (auxiliary feedback) for each order. Therefore, our proposed method uses all observations with reward samples to estimate the reward function $f$.
>
> In practice, one can use an appropriate variant of our proposed method depending on the need of their application (EH and BE variants shown in Fig. 2(a)-2(c)). However, it will be challenging to quantify the exact gain in variance reduction (as shown in Theorem 3 for specific case).
>
>
> > ### *The IS and MF sampling strategy are only listed in the method discussions, and for the experiments when $q=2$, it is hard to compare the pros and cons of them as they are equivalently being the same. Could the authors add more results for larger $q$ to showcase the effectiveness of the two sampling methods. A larger $q$ might also be helpful in understanding the effectiveness of having more control variates.*
>
> In control variate literature, it has been proven that IS and MF sampling strategies are asymptotically optimal (Gorodetsky et al., 2020), i.e., the variance reduction achieved by both strategies is asymptotically the same as if auxiliary feedback functions are known. Further, it is also shown that they both have similar empirical performance (Gorodetsky et al., 2020, Fig. 4), but these results are shown for a non-parametric offline setting. However, both strategies can be used for different problems: IS sampling strategy suits the problems in which different auxiliary feedback can be independently sampled, whereas MF sampling suits problems where auxiliary feedback can not be sampled independently. We will add experiments on both sampling strategies with larger $q$ in the future version of the paper. When the auxiliary functions are known, we have already added an experiment result with larger $q (={1, 2, 3, 4, 5}$). For more details, please check the attached pdf in the global rebuttal.
>
>
> > ### *I am not sure if there is any public dataset available to showcase the effectiveness of the method in more real-world scenarios. The current ablation experiments facilitate the understanding of the method, but adding real-world experiments would be much more convincing.*
>
> The main goal of this paper is to propose a method that exploits the auxiliary feedback correlated with reward samples to improve the performance of parameterized bandit algorithms and quantify performance gain (i.e., reduction in regret). It is common in bandit literature to measure the performance of bandits algorithms on synthetically generated data. We have also validated our method using several bandits instances.
>
> In the future, we will set up an elaborate experiment to demonstrate the effectiveness of our algorithms on real datasets.
>
>
> > ### *For Figure 2 (f), is it possible to add the performance for original Lin-UCB performance?*
>
> We have added the Lin-UCB in Figure 2(f). For more details, please check Figure 1 in the attached pdf in the global rebuttal.
>
>
> > ### *It would be good to have an ablation study w.r.t the number of auxiliary functions.*
>
> We have added an experiment result with more auxiliary functions ($q={1, 2, 3, 4, 5}$). For more details, please check Figure 2 in the attached pdf in the global rebuttal.

---

> > ### Comment · Reviewer_CBCn · 2023-08-15
> > **Thank you for the response.**
> >
> > Thanks for the authors' response, and I believe my initial score appropriately reflects the quality of this work.

---

> > > ### Author Response · Authors · 2023-08-16
> > > **Thank you for your review**
> > >
> > > Dear Reviewer CBCn,
> > >
> > > Thank you for acknowledging our response and maintaining positive opinion of our work. Your feedback are tremendously valuable to us, and we will include our responses in the revised version to further improve our paper.
> > >
> > > Regards,\
> > > Authors

---

### Official Review · Reviewer_vUBt · 2023-07-05

**Soundness:** 4 excellent
**Presentation:** 3 good
**Contribution:** 3 good
**Rating:** 6
**Confidence:** 3

**Summary:**

This paper focus on the problem of parameterized bandits when extra auxiiliary feedback is available, which can be utilized to construct an unbiased reward estimator, which potentially shares smaller variance and thus algorithms based on such estimator can thus incur smaller regret. Experiments validates the estimator and corresponding algorithm.

**Strengths:**

- The paper is clearly-written, in which the problem statement, algorithms, theories and experiments are easy to follow.
- Control variate theory is applied in this paper, which is suitable for i.i.d. random process in the variance reduction, which is one of the central topic in bandits, and is of independent interests beyond the topic.
- The proposed algorithm is evaluated both empirically and theoretically.

**Weaknesses:**

- The improvement is expectable, as auxiliary feedback (AF) requires more information and computation. So this can be seen as a trade-off between extra information beyond rewards, extra computation and a constant order improvement in regret.
- The hybrid reward only reduces variance when the AF is correlated with reward with same covariance $\sigma_{y,w}$. I'm worried this could not be true for most real-world applications.
- Though extra AF is given, the experiments does not design how OFUL or LinUCB can utilize such AFs. One can figure another simplest way to use these extra information, for example, learn a model $f:\mathbf{R}^d \rightarrow \mathbf{R}$, which means to learn the function that maps the AF vector to the reward $y_t$, and thus OFUL and LinUCB can gain extra feedback $f(AF)$ to better estimate the reward. If the baseline just discard the extra AF, its' not fair.

**Questions:**

See weaknesses above.

---

> ### Author Rebuttal · Authors · 2023-08-10
>
> Thank you for your insightful comments. In the following, we have responded to your questions.
>
>
> > ### *The improvement is expectable, as auxiliary feedback (AF) requires more information and computation. So this can be seen as a trade-off between extra information beyond rewards, extra computation and a constant order improvement in regret.*
>
> We agree with the reviewer's observation that there is a trade-off between extra computations needed to incorporate auxiliary feedback in existing bandit algorithms and improvement in regret. However, regret improvement is only possible when the auxiliary feedback correlate with the reward, and the regret improvement is upper bounded by the correlation between the reward and its auxiliary feedback.
>
>
> > ### *The hybrid reward only reduces variance when the AF is correlated with reward with same covariance $\sigma_{y,w}$. I'm worried this could not be true for most real-world applications.*
>
> Yes, we assumed that the correlation between reward and its auxiliary is the same across all actions, i.e., same covariance $\sigma_{y,w}$ (Line 101). This assumption is reasonable as the source of randomness in reward and auxiliary feedback is zero mean Gaussian noise, whose variance is independent of the action (it is a common assumption in many bandit algorithms like OFUL, Lin-UCB, and UCB-GLM). However, this assumption can be violated in many real-world applications where Gaussian noise varies across actions, making covariance $\sigma_{y,w}$ vary across actions. The closest bandit setting to this problem is the bandit problem with heteroscedastic noise. Even though the definition of hybrid rewards used in the paper can be used, it may not give the best possible variance reduction. Therefore, the problem of varying covariance $\sigma_{y,w}$ needs to be systematically studied and can be an independent work, as we mentioned in Line 344.
>
>
>
> > ### *Though extra AF is given, the experiments does not design how OFUL or LinUCB can utilize such AFs. One can figure another simplest way to use these extra information, for example, learn a model $f: R^d \rightarrow R$, which means to learn the function that maps the AF vector to the reward $y_t$, and thus OFUL and LinUCB can gain extra feedback $f(AF)$ to better estimate the reward. If the baseline just discard the extra AF, its' not fair.*
>
> We do not know any parameterized bandit algorithm that can exploit the available auxiliary feedback. Therefore, our goal is to design a method that can exploit correlated auxiliary feedback to improve the performance (i.e., minimize regret) of the existing bandit algorithms. To achieve that, we use auxiliary feedback as control variates and extend the existing results from the control variate theory to our setting. We use vanilla OFUL and Lin-UCB as baselines to demonstrate the performance gain achieved by our approach.
>
> Let $f: R^d \rightarrow R$ learn the function that maps the auxiliary feedback vector to the reward $y_t$. Then, extra feedback $f(AF)$ may not get a better reward estimate in OFUL and LinUCB. Because first, the auxiliary feedback vector may only partially correlate with reward, e.g., user rating of food also depends on food taste and quality (which can not be observed by the platform) apart from the food delivery time. Second, getting an estimate for the reward from auxiliary feedback is impossible as auxiliary feedback is only observed with the reward.

---

> > ### Comment · Reviewer_vUBt · 2023-08-16
> >
> > Thanks for clarifying and authors clearly address my concerns, and I tend to keep my score. Good luck with the final decision.

---

> > > ### Author Response · Authors · 2023-08-16
> > > **Thank you for your review**
> > >
> > > Dear Reviewer vUBt,
> > >
> > > Thank you for your positive feedback. We are glad that we were able to address all your concerns. We will include all our responses in the revised version of the paper.
> > >
> > > Regards,\
> > > Authors

---

### Author Rebuttal · Authors · 2023-08-10

We thank all reviewers for their time and efforts in evaluating our paper and for their detailed comments and suggestions. We hope our answers to your questions will alleviate your concerns and further improve your opinion of our work. If you have additional questions, we would be happy to address them.

Here, we address two main concerns and respond to your questions in individual rebuttals.


### **Main contributions:**
The following are our main contributions:
- **General setup:** Verma and Hanawal (2021) focus on a non-parameterized bandit setting, which assumes a finite number of actions and known auxiliary mean values. In contrast, we consider a more general bandit setting with a large (or even infinite) number of actions (i.e., parameterized bandits with contextual information). Further, we extend to a setting where unknown functions parameterize different auxiliary feedback.

- **Control variates theory with parameterized function:** Control variate literature focuses on a non-parameterized control variate (auxiliary feedback in our problem), i.e., control variates are sampled from a fixed distribution. We first extend the existing control theory results to the problems where known functions parameterize the control variates (Section 3) and then extend to problems where functions parameterizing control variates are unknown (Section 4). Our key contribution is to design an unbiased reward function estimator using hybrid rewards (a combination of reward and its auxiliary feedback), which gives a maximum reduction in the estimator's variance. These contributions are themselves of independent interest in control variate theory.

- **AFC bandit algorithm:** We introduce the notion of the Auxiliary Feedback Compatible (AFC) bandit algorithm. A bandit algorithm is an AFC bandit algorithm when certain conditions are satisfied (more details are in Definition 1). One can use hybrid rewards instead of only observed rewards in the AFC bandit algorithm, which leads to tighter confidence bounds and hence smaller regret.

Our work has shown that the regret of AFC bandit algorithms can be improved by exploiting the auxiliary feedback. We hope the proposed method and techniques can be used for more challenging bandit problems with auxiliary feedback, e.g., bandit problems with heteroscedastic noise, non-Gaussian noise, adversarial contexts, and different environments.


### **Availability of Auxiliary feedback**
Auxiliary feedback is easily available in many real-life applications. To illustrate that, we consider following different scenarios:
- **Reward sample with auxiliary feedback:** In many problems, reward and its auxiliary feedback are observed jointly in each round. For example, consider a job schedular (Verma and Hanawal, 2021) that aims to assign different jobs to available servers. The job's service time (reward) depends on its size (auxiliary feedback) and other factors (e.g., load in assigned server).
In such settings, reward and auxiliary feedback are observed jointly. Therefore, the platform can either use available historical data to estimate the mean job size or use observed auxiliary feedback to estimate auxiliary functions but ignore the associated reward samples (this is inefficient).


- **Auxiliary feedback with no reward sample:** Consider an online food delivery platform that keeps track of the user's rating (reward) to recommend the best-rated restaurant and can also observe food delivery time (auxiliary feedback) for each order. The platform can observe the delivery time for every order but may only sometimes receive user ratings. In such settings, additional auxiliary feedback is available apart from historical data, which can be used for estimating the auxiliary feedback functions. Similar scenarios arise in online cab booking platforms and e-commerce platforms, as mentioned in the paper.


- **Sampling auxiliary feedback without reward sample:**
Consider a problem where getting samples from the high-fidelity simulation is very expensive but accurate. However, cheap low-fidelity simulations are available and correlated with expensive high-fidelity simulations. In such scenarios, one can independently collect sufficient samples to get a good estimate of low-fidelity simulations, and then the samples from cheap low-fidelity simulations can be used to minimize the variance of the estimator based on high-fidelity simulations.

---

### Decision · Program_Chairs · 2023-09-21

**Decision:**

Accept (poster)

**Comment:**

This paper generalizes multi-armed bandits with control variates to linear bandits. The proposed algorithm is a variant of LinUCB and computationally efficient. Its analysis shows lower regret due to control variates. The algorithm is empirically evaluated on several synthetic problems.

This paper is easy to read and sound. While the extension is novel, it is not surprising. The additional shortcomings are:

* One strong assumption in this work (also in prior works) is that the auxiliary feedback functions are known. This is rarely true in practice. To counter this point, I suggest that the authors add experiments where the auxiliary feedback functions are estimated from historical logged data. Similar experiments have become common in recent works on hierarchical Bayesian bandits, such as [Mixed-Effect Thompson Sampling](https://proceedings.mlr.press/v206/aouali23a.html), where the model is often unknown and has to be estimated from historical logged data. No analysis is needed. This is for practitioners.

* The proposed algorithm is not compared to any naive way of incorporating side information, such as adding it as features. While this is not theoretically sound, it is a leading approach in practice. Showing that this approach can be beaten would convince practitioners to pay attention to this work.

While this work has shortcomings, they can be addressed with a little bit more work and I hope that the authors will do so. With this, I suggest acceptance of this paper.